# LncRNA *DANA1* promotes drought tolerance and histone deacetylation of drought responsive genes in *Arabidopsis*

Jingjing Cai[1,5], Yongdi Zhang[1,5], Reqing He[1,5], Liyun Jiang [ID] [1,5], Zhipeng Qu[2], Jinbao Gu[3], Jun Yang[1], María Florencia Legascue[4], Zhen-Yu Wang[3], Federico Ariel[4], David L Adelson [ID] [2], Youlin Zhu[1] & Dong Wang [ID] [1✉]

## Abstract

**Although many long noncoding RNAs have been discovered in plants, little is known about their biological function and mode of action. Here we show that the drought-induced long intergenic noncoding RNA *DANA1* interacts with the L1p/L10e family member protein *DANA1*-INTERACTING PROTEIN 1 (DIP1) in the cell nucleus of Arabidopsis, and both *DANA1* and DIP1 promote plant drought resistance. *DANA1* and DIP1 increase histone deacetylase HDA9 binding to the *CYP707A1* and *CYP707A2* loci. DIP1 further interacts with PWWP3, a member of the PEAT complex that associates with HDA9 and has histone deacetylase activity. Mutation of *DANA1* enhances *CYP707A1* and *CYP707A2* acetylation and expression resulting in impaired drought tolerance, in agreement with *dip1* and *pwwp3* mutant phenotypes. Our results demonstrate that *DANA1* is a positive regulator of drought response and that *DANA1* works jointly with the novel chromatin-related factor DIP1 on epigenetic reprogramming of the plant transcriptome during the response to drought.**

**Keywords** Long Noncoding RNA; RNA-Binding Protein; Histone Deacetylation; Drought Response
**Subject Categories** Chromatin, Transcription & Genomics; Plant Biology; RNA Biology

## Introduction

With the advent of powerful next-generation sequencing technologies, tens of thousands of long noncoding RNAs (lncRNAs) have been identified in eukaryotic transcriptomes (Palazzo and Koonin, 2020; Rinn and Chang, 2020). Those over 200-nt noncoding transcripts have emerged as important players of gene regulation at transcriptional, post-transcriptional, translational and post-translational levels (Statello et al, 2021; Yao et al, 2019). In plants, lncRNAs have been implicated in a wide range of developmental processes, including seedling photomorphogenesis, flowering, reproduction, fruit ripening, and stress response (Ariel et al, 2015; Ben Amor et al, 2009; Ding et al, 2012; Fang et al, 2019; Li et al, 2018; Seo et al, 2019; Wang et al, 2017; Wang et al, 2014; Wang et al, 2018; Zhang et al, 2014; Zhao et al, 2018; Zheng et al, 2019; Zhou et al, 2012). Although a large number of lncRNAs have been identified in diverse plant species, only a handful of them have been experimentally characterized. Among them, lncRNA interacting proteins emerged as critical factors in the activity of the noncoding transcriptome (Lucero et al, 2021). Among other molecular mechanisms, lncRNA-integrated ribonucleoprotein complexes participate in the epigenetic regulation of gene expression. For example, *COOLAIR*, a group of long antisense RNAs expressed from the *FLOWERING LOCUS C* (*FLC*) locus, represses *FLC* transcription with increased H3K27me3 and decreased H3K36me3 levels (Csorba et al, 2014), and the RNA-binding protein FCA interacts with the PRC2 subunit CURLY LEAF (CLF) and binds *COOLAIR* transcripts to allow deposition of H3K27me3 at *FLC* (Tian et al, 2019). Unlike *COOLAIR*, both *COLDAIR* and *COLDWARP* are lncRNAs transcribed in a sense direction relative to *FLC* mRNA, and directly associate with PRC2 to suppress *FLC* expression (Heo and Sung, 2011; Kim and Sung, 2017), similar to *AG-incRNA4* that mediates the CLF-dependent deposition of H3K27me3 in the first intron of *AGAMOUS* (Wu et al, 2018). In addition, the Trithorax H3K4 methyltransferase ARABIDOPSIS TRITHORAX-LIKE PROTEIN 1 (ATX1), mediating the establishment of H3K4me3 (Alvarez-Venegas et al, 2003), is also modulated by *COLDAIR* (Liu et al, 2020). This histone mark is also regulated by the lncRNAs *MADS AFFECTING FLOWERING4* (*MAS*) in *Arabidopsis* and *LRK Antisense Intergenic RNA* (*LAIR*) in rice through the modulation of the COMPASS-like complex component WDR5a (Wang et al, 2018; Zhao et al, 2018). Notably,

[1]Key Laboratory of Molecular Biology and Gene Engineering in Jiangxi Province, College of Life Science, Nanchang University, 330031 Jiangxi, China. [2]Department of Molecular and Biomedical Science, School of Biological Sciences, The University of Adelaide, Adelaide 5005 SA, Australia. [3]Institute of Nanfan & Seed Industry, Guangdong Academy of Sciences, 510316 Guangdong, China. [4]Instituto de Agrobiotecnología del Litoral, CONICET, FBCB, Universidad Nacional del Litoral, Colectora Ruta Nacional 168 km 0, Santa Fe 3000, Argentina. [5]These authors contributed equally: Jingjing Cai, Yongdi Zhang, Reqing He, Liyun Jiang. ✉E-mail: dongwang@ncu.edu.cn

*LAIR* also interacts with the H4K16 acetyltransferase OsMOF (Wang et al, 2018), suggesting that lncRNAs are central modulators of chromatin dynamics. The ELF18-induced lncRNA *ELENA1* enhances *PR1* transcription by cooperating directly with Mediator subunit 19a (MED19a) and influencing its enrichment on the *PR1* promoter (Seo et al, 2017). The long intergenic noncoding RNA (lincRNA) *APOLO*, not only interacts with the PRC1 subunit LIKE HETEROCHROMATIN PROTEIN 1 (LHP1) to form a chromatin loop encompassing the promoter of its neighboring gene *PID* and downregulate the transcription of *PID* (Ariel et al, 2014), but also coordinates expression of distal unrelated auxin-responsive genes through sequence complementarity and R-loop formation (Ariel et al, 2020). Interestingly, it has been recently shown that LHP1 also recognizes in vivo the lncRNA *MARS*, involved in the transcriptional regulation of a non-homologous gene cluster in *Arabidopsis* (Roulé et al, 2022).

Drought is an environmental stress that can severely limit plant productivity. Plants respond to drought by altering their metabolism, physiology and development, which could mitigate the negative impact of water deficit on their growth and reproduction (Gupta et al, 2020). The phytohormone abscisic acid (ABA) is the major signaling molecule in plant responses to drought stress. ABA content increases when a land plant is subjected to drought stress, and it rapidly decreases when the land plant is recovering from drought. ABA content in plants is modulated by the balance between its biosynthesis and catabolism, and ABA is catabolized via two pathways, hydroxylation and glucose conjugation (Cutler and Krochko, 1999; Nambara and Marion-Poll, 2005). The 8'-hydroxylation is thought to be the predominant pathway for ABA catabolism, which is catalyzed by a cytochrome P450 monooxygenase encoded by *CYP707As* (Kushiro et al, 2004; Saito et al, 2004). Protein-encoding genes involved in drought response have been widely studied, but little is known about the role of lncRNA regulation of drought responses in plants. In potato, *CYCLING DOF FACTOR 1* (*StCDF1*) together with its lncRNA counterpart, *StFLORE*, regulates drought response (Ramírez Gonzales et al, 2021). The rice lncRNA TCONS_00021861 can regulate *YUCCA7* expression by sponging miR528-3p, which in turn activates the indoleacetic acid (IAA) biosynthetic pathway and confers resistance to drought stress (Chen et al, 2021). A maize lncRNA, *cis-NATZmNAC48* together with *ZmNAC48* form natural antisense transcripts-generated small interfering RNA (nat-siRNA) through double-stranded RNA-dependent mechanisms, and *cis-NATZmNAC48* overexpression in maize results in a higher water-loss rate and dead leaves (Mao et al, 2021). Overexpressing the lncRNA *DRIR* makes *Arabidopsis* more resistant to drought than the wild-type (WT) (Qin et al, 2017), nevertheless, the molecular basis of its function remains unknown.

The PEAT complex, composed of four proteins belonging to the respective PWWP, EPCR, ARID, and TRB families, is not only involved in heterochromatin formation and gene repression through histone deacetylation but also appears to have a locus-specific activating role, possibly through promoting histone acetylation (Tan et al, 2018; Tsuzuki and Wierzbicki, 2018). As a member of the PEAT complex, PWWP3 loss of function mutant has no obvious developmental defects (Tan et al, 2018), and its role in plant drought response is still unclear.

In this study, we have identified a lncRNA, *DANA1* (for *Drought Associated long Noncoding RNA 1*), which acts as a new positive

regulator of drought response, according to the behavior of *DANA1* knockout and overexpressing plants. Genome-wide transcriptome analysis of *dana1* mutants showed elevated expression of drought response-related genes, including *CYP707A1* and *CYP707A2*. Transcriptional regulation of *CYP707A1* and *CYP707A2* by *DANA1* was brought about by the interaction with DIP1 (for *DANA1*-interacting protein 1) annotated as a ribosomal protein L1p/L10e family protein, leading to the alternations of H3K9ac and H3K27ac enrichment on both loci. *DANA1* together with DIP1 can influence the HDA9 binding on *CYP707A1* and *CYP707A2* loci by means of PWWP3 that is a member of the PEAT complex, involved in histone deacetylation. Taken together, our results demonstrate that *DANA1* works jointly with DIP1 to regulate the HDA9 enrichment on chromatin of the *CYP707A1* and *CYP707A2* loci, thereby modulating *CYP707A1* and *CYP707A2* expression during the drought response in plants.

## Results

### The lncRNA *DANA1* is involved in *Arabidopsis* response to drought

To investigate functional roles of lncRNAs in plant drought response, we identified homozygous T-DNA insertion mutants in a number of lncRNAs reported in our previous study (Wang et al, 2017). PEG treatment, which could impose a low water potential that is reflective of the type of stress imposed by a drying soil (Verslues et al, 2006), was chosen to perform a rapid and efficient screening of mutants. Strikingly, two independent T-DNA insertion alleles of *DANA1* exhibited PEG-sensitive phenotypes (Fig. 1A–D). T-DNA insertions in both mutants caused *DANA1* transcript to be undetectable (Fig. 1B), and we designated these two knockout lines as *dana1-1* and *-2*. *DANA1* is a single-exon noncoding transcript of 786 nucleotides (previously known as CUFF.13633 (Wang et al, 2017)), located in the intergenic region between AT4G14540 (*NF-YB3*) and AT4G14548 (Fig. 1A). Interestingly, there is another lncRNA (AT4G06195) annotated in TAIR10 just inside *DANA1* locus with the opposite transcriptional direction (Appendix Fig. S1A). We then used strand-specific RT-PCR to detect its transcripts and found that *DANA1* transcripts can only be detected in this locus (Appendix Fig. S1B). Data from Transcription Start Site sequencing (Data ref: Nielsen et al, 2019) also supported our strand-specific RT-PCR result (Appendix Fig. S1A), further indicating that *DANA1* RNAs is the only detectable transcript in this locus. Under normal growth condition, *dana1-1* and *dana1-2* seedlings had similar growth when compared to WT (Fig. 1C). However, when grown on agar plates supplemented 20% (w/v) PEG, slightly but significantly increased sensitivity was exhibited by the mutants compared with WT (Fig. 1C). Both mutants exhibited a significant reduction of primary root elongation and fresh weight of aerial tissues when compared to WT under PEG treatment (Fig. 1D, $P < 0.05$, Student's *t* test). In addition, the expression level of *DANA1* in dehydrated seedlings was more than twofold that in untreated seedlings (Appendix Fig. S1C), indicating that drought evidently up-regulates the expression of *DANA1*. Our RT-qPCR assays showed that *DANA1* transcript levels were strongly increased in rosette leaves in response to drought stress treatment, peaking at the 6th day of drought treatment (Fig. 1E). Next, in order to uncover the role of *DANA1* in plant responses to drought, we

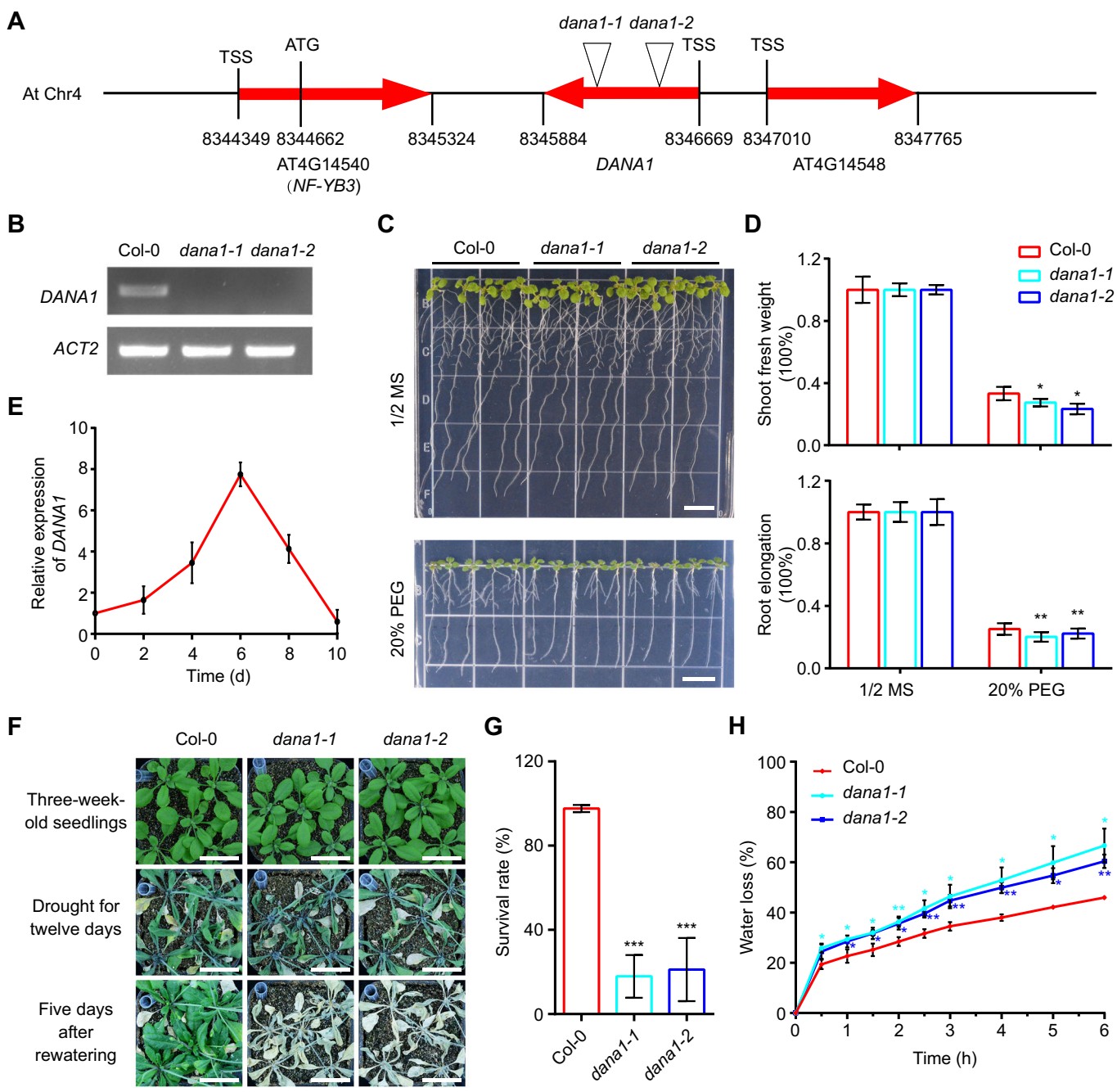

**Figure 1. DANA1 mutants are more sensitive to drought stress.**

(A) Schematic representation of the *DANA1* locus. T-DNA insertions are marked with triangles. TSS, transcription start site; ATG, translation start site. (B) Expression analysis of *DANA1* in wild-type (Col-0) and two T-DNA insertion mutant alleles (*dana1-1* and *dana1-2*). (C) *DANA1* mutants are sensitive to polyethylene glycol 8000 (PEG) treatment. Scale bars = 1 cm. (D) Root length and fresh weight of seedlings shown in (C). Both graphs are presented as the percentage relative to growth on control half-strength MS medium. Error bars represent standard deviation (*n* = 15, *n* refers to biological replicates). (E) Quantitative measurement of the *DANA1* transcript levels in 3-week-old Col-0 plants after drought stress for 0, 2, 4, 6, 8, or 10 days. *UBQ3* was used as an internal control. (F) Morphology of seedlings before and after drought stress treatment. Three-week-old Col-0, *dana1-1* and *dana1-2* plants were subjected to drought stress for 12 days and then rewatered for 5 days. Scale bars = 3 cm. (G) Survival rate after drought treatment (*n* = 3 biological replicates, *n* refers to biological replicates). (H) Water loss in detached leaves of 3-week-old Col-0, *dana1-1* and *dana1-2* plants (*n* = 3, *n* refers to biological replicates, each replicate contains five fully expanded leaves). Data information: Values shown are means ± SD from three biological replicates. Asterisks represent significant differences determined by Student's *t* test (*$P < 0.05$; **$P < 0.01$; ***$P < 0.001$). Source data are available online for this figure.

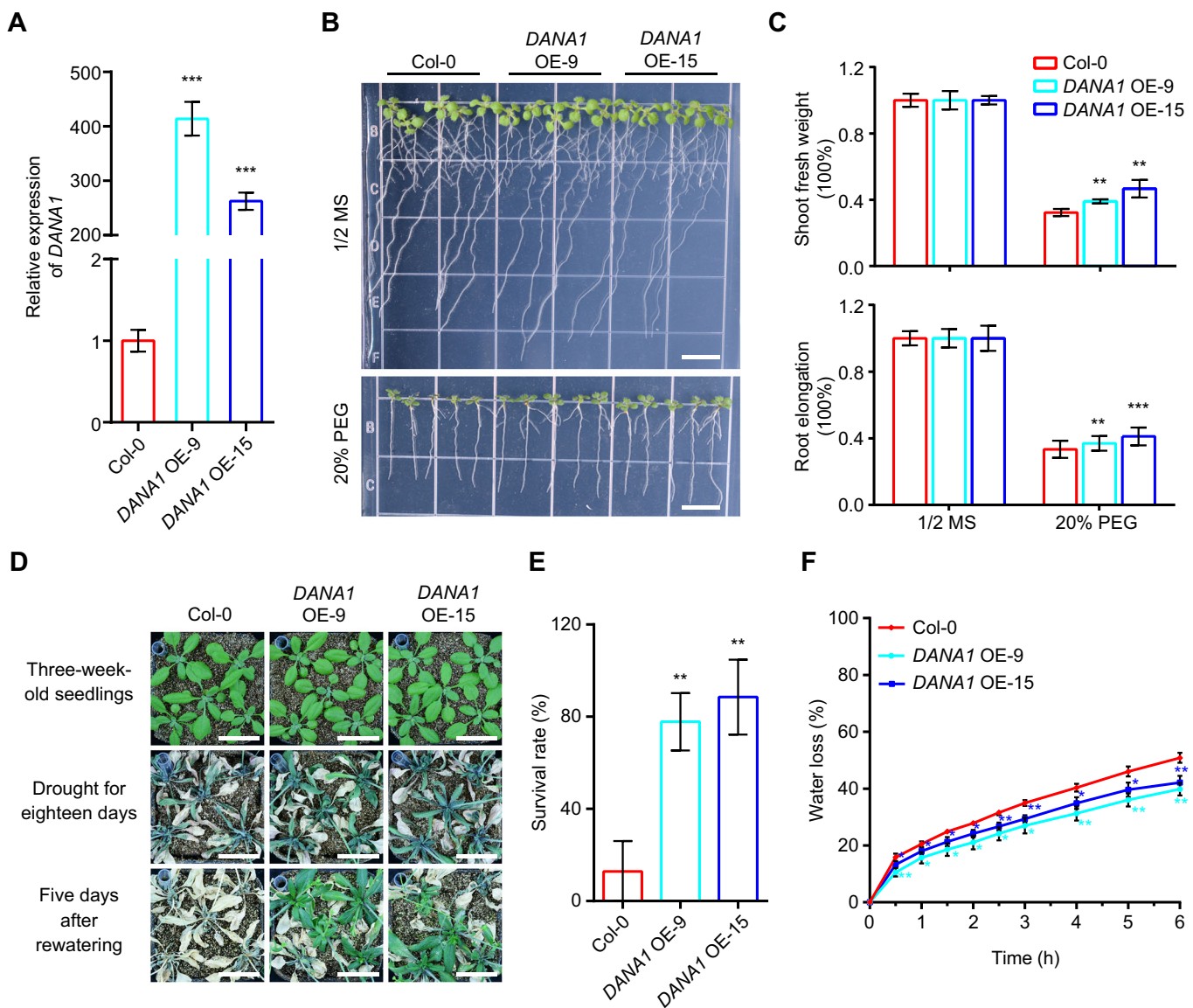

**Figure 2. Drought-tolerant phenotype of *DANA1* overexpressing plants.**

(A) Relative transcript levels of *DANA1* in Col-0 and two overexpressing lines (35 S: *DANA1* in Col-0, *DANA1* OE-9 and *DANA1* OE-15). (B) The *DANA1* overexpressing lines are insensitive to PEG treatment. Scale bars = 1 cm. (C) Root length and fresh weight of seedlings shown in (B). Error bars represent standard deviation (n = 15, n refers to biological replicates). (D) Drought-tolerance assay. Col-0, *DANA1* OE-9 and *DANA1* OE-15 plants grown under normal growth conditions for 3 weeks were subjected to drought stress for 18 days and then rewatered for 5 days. Scale bars = 3 cm. (E) Survival rate after drought treatment (n = 3, n refers to biological replicates). Scale bars = 3 cm. (F) Water loss in detached leaves of 3-week-old Col-0, *DANA1* OE-9, and *DANA1* OE-15 plants. Data information: Values shown are means ± SD from three biological replicates. Asterisks represent significant differences determined by Student's *t* test (*$P < 0.05$; **$P < 0.01$; ***$P < 0.001$). Source data are available online for this figure.

phenotyped *dana1-1* and *dana1-2* under drought stress. Both independent mutants were substantially more sensitive to drought stress than WT plants (Fig. 1F). Compared with a WT survival rate of ~97%, only ~17.9% of *dana1-1* and ~21.1% of *dana1-2* mutant plants survived after twelve days of drought stress followed by a 5-day recovery period (Fig. 1G). Water-loss analysis showed that detached leaves of *dana1* mutants lost water more quickly than did those of Col-0 at multiple time points (Fig. 1H), consistent with the drought-hypersensitive phenotype of *dana1* mutants. Based on increased water loss in the *dana1* mutant, we examined the stomatal indexes from the abaxial epidermis of WT, *dana1-1* and

*dana1-2* plant leaves. The loss of *DANA1* function induced an increase in the stomatal aperture index (Appendix Fig. S2A). Moreover, *dana1* mutants failed to exhibit ABA-induced stomatal closure to the same extent as WT (Appendix Fig. S2B). Considering that *NF-YB3* is located approximately 600 bp downstream of *DANA1* and is able to regulate the expression of dehydration stress-inducible genes (Sato et al, 2019), we examined the expression level of *NF-YB3* in *dana1* mutants. The RT-qPCR results showed that mutated *DANA1* did not influence *NF-YB3* expression (Appendix Fig. S3A). In addition, *NF-YB3* expression level in the *DANA1* complementation line (*DANA1* COM) plants was similar to Col-0

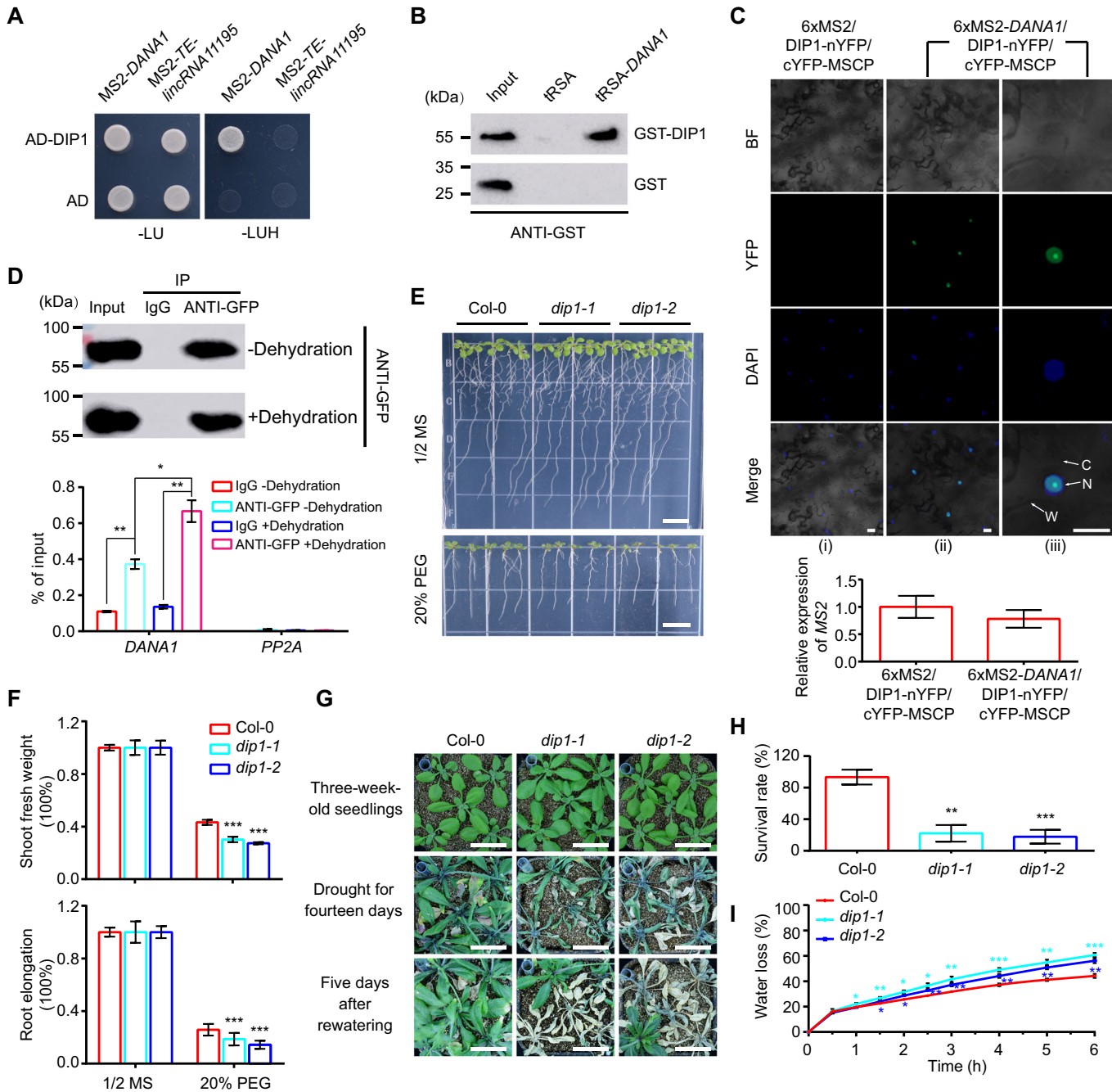

(Appendix Fig. S3B,C). To further verify the drought-sensitive phenotype caused by loss of *DANA1*, the *DANA1* COM was tested for drought tolerance. The drought-tolerance assay showed the same phenotype as Col-0 (Appendix Fig. S3D,E). Moreover, both *DANA1* COM and Col-0 exhibited comparable water-loss rate, stomatal indices and ABA-induced stomatal closure (Appendix Figs. S2C,D and S3F). Altogether, our results indicate that *DANA1* mediates stomata closure of *Arabidopsis* plants in response to drought.

The role of *DANA1* in drought stress response was further evaluated by overexpressing *DANA1* in plants, a construct consisting of the cauliflower mosaic virus 35 S promoter was

transferred to Col-0, and two independent *DANA1* overexpressing lines (*DANA1* OE-9 and *DANA1* OE-15) were obtained and subjected to PEG treatment (Fig. 2A). Both *DANA1* overexpressing lines presented markedly enhanced tolerance when compared to WT under PEG treatment (Fig. 2B,C). Then, 3-week-old seedlings of Col-0 and two *DANA1*-overexpressing lines were treated with drought stress and rewatering. The drought stress assay indicated that both *DANA1* OE-9 and *DANA1* OE-15 were more tolerant to drought stress than the Col-0 (Fig. 2D). Compared with a ~12% survival rate in the WT, up to ~77% of *DANA1* OE-9 and ~88% of *DANA1* OE-15 transgenic plants survived after 18 days of drought stress followed by a 5-day recovery period (Fig. 2E). Consistent

**Figure 3. *DANA1* interacts with DIP1.**

(A) Tests of *DANA1*-DIP1 interaction by yeast three-hybrid assays. Full-length DIP1 protein was fused with the GAL4 activation domain (AD). MS2 nucleotide sequences fused to *DANA1* or *TE-lincRNA11195*. (B) In vitro binding assay with in vitro transcribed *DANA1* fused with tRSA (the addition of the tRNA scaffold to a Streptavidin aptamer) and recombinant glutathione S-transferase (GST)-DIP1 protein. The in vitro transcribed tRSA was used as a negative control. (C) Examination of interaction between *DANA1* and DIP1 by trimolecular fluorescence complementation (TriFC) assay in *Nicotiana benthamiana* leaves. The N-terminal fragment of yellow fluorescent protein (nYFP) was fused to DIP1, and C-terminal fragment of yellow fluorescent protein (cYFP) to MS2 coat protein (MSCP). 6xMS2 nucleotide sequences were fused with or without *DANA1*. (Ci) is the negative control, and (Ciii) is the magnified image of (Cii). Confocal images were taken 2 days after infiltration. BF bright field, C cytoplasm, N nucleus, W cell wall. Scale bars = 20 µm. Expression levels of 6xMS2 and 6xMS2-*DANA1* in leaves of tobacco transiently expressed corresponding vectors were examined on the bottom panel. (D) RIP assay with *UBQ10:DIP1-GFP* plants during dehydration treatment. Purification of DIP1 was validated by western blot (upper panel), and the levels of *DANA1* in the immunoprecipitates were determined by RT-qPCR (lower panel). IgG was used as the negative control, and the RNA *PP2A* was used as a RIP negative control in *Arabidopsis* plants. Ten-day-old seedlings were either dehydrated to lose ~60% fresh weight or not (-Dehydration). (E) *DIP1* mutants are sensitive to PEG treatment. Scale bars = 1 cm. (F) Root length and fresh weight of seedlings shown in (E). Error bars represent standard deviation (n = 15, n refers to biological replicates). (G) Drought-tolerance assay. Col-0, *dip1-1* and *dip1-2* plants grown under normal growth conditions for 3 weeks were subjected to drought stress for 14 days and then rewatered for 5 days. Scale bars = 3 cm. (H) Survival rate after drought treatment (n = 3, n refers to biological replicates). (I) Water loss in detached leaves of 3-week-old Col-0, *dip1-1* and *dip1-2* plants. Data information: Values shown are means ± SD from three biological replicates. Asterisks represent significant differences determined by Student's t test (*$P < 0.05$; **$P < 0.01$; ***$P < 0.001$). Source data are available online for this figure.

with the drought-tolerance phenotype, detached leaves from *DANA1*-overexpressing plants exhibited reduced water loss (Fig. 2F). Stomata of *DANA1* overexpressing plants had smaller aperture than did Col-0 plants (Appendix Fig S2E,F). With regard to that *DANA1* OE plants with 200–400-fold increase exhibited mild phenotype, which is reasonable because the lncRNA *DRIR* overexpression line with the 500–600-fold increase also has a similar drought phenotype and water-loss rate to the lncRNA *DRIR* overexpression line with an ~40-fold increase (Qin et al, 2017). Together, these results further support that *DANA1* is an important positive regulator of drought tolerance.

## *DANA1* regulates the expression of genes involved in the drought stress response

To elucidate the molecular mechanism involving *DANA1* in the plant response to drought stress, we performed RNA-seq on 10-day-old WT and *dana1-2* seedlings. Compared with WT, there were 634 significantly upregulated genes and 1140 significantly downregulated genes in *dana1-2* (Appendix Fig. S4A and Dataset EV1). Gene Ontology (GO) enrichment analysis for the biological process of the differentially expressed genes (DEGs) indicated that genes involved in biological processes of "Response to ABA", "Response to hormone", "Response to abiotic stimulus", "Defense response" and so on were significantly over-represented (Dataset EV2). The genomic distribution of these 100 most significant DEGs pointed out that they are distributed across all chromosomes (Appendix Fig. S4B). More importantly, a series of drought-responsive genes have been found to exhibit the most significantly different expression changes (see Dataset EV1), and we randomly selected six of these DEGs for validation of the RNA-seq results by RT-qPCR. We confirmed the upregulation in *dana1* mutant of ABA catabolic genes, including *CYP707A1*, *CYP707A2*, *CYP707A4*, and *UGT71B1*, as well as the downregulation of *MYB44* and *NTL6* (Appendix Fig. S4C). Altogether, our results indicate that the deregulation of *DANA1* has an impact on the plant transcriptome, notably of drought-responsive genes, in agreement with the physiological role of *DANA1*.

## *DANA1* interacts with DIP1

Considering that *DANA1* overexpressing plants exhibited an opposite response compared to the *dana1* mutant with respect to

PEG and drought phenotypes (Figs. 1C,F and 2B,D), we hypothesized that *DANA1* regulates the expression of genes associated with plant drought response in trans. The function of lncRNAs is primarily determined by the ribonucleoprotein complexes they are part of (Lucero et al, 2021). Thus, in order to identify *DANA1*-protein partners, we generated a yeast three-hybrid cDNA library of 410 genes that are able to bind to RNAs in *Arabidopsis thaliana* (Köster et al, 2017), to screen for *DANA1*-interacting proteins (Appendix Table S2). As a result, we identified the gene AT1G06380, which contains a ribosomal domain, belonging to the L1p/L10e family (hereafter called *DANA1*-INTERACTING PROTEIN 1, DIP1; Fig. 3A). To validate this interaction, we fused a tRNA scaffold containing a Streptavidin aptamer (tRSA) (Iioka et al, 2011) with full-length *DANA1*, and we found that DIP1 specifically bound to tRSA-*DANA1* but not to tRSA alone (Fig. 3B). We then examined in vivo association of *DANA1* with DIP1 using a trimolecular fluorescence complementation (TriFC) assay (Seo et al, 2017), and observed that *DANA1* associated with DIP1 in the nucleus (Fig. 3C), which is also supported by the subcellular localization of *DANA1* predominantly in the nucleus (Appendix Fig. S5), hinting at a ribosome-independent role of the *DANA1*-DIP1 complex, in spite of the primary classification and annotation of this gene (Krishnakumar et al, 2015). Finally, in vivo association of *DANA1* with DIP1 was further confirmed by RNA immunoprecipitation (RIP) assay in the *UBQ10: DIP1-GFP* transgenic lines (Appendix Fig. S6A,B) where DIP1 pulled down endogenous *DANA1* (Fig. 3D); more importantly, there was increased association of *DANA1* with DIP1 during dehydration treatment (Fig. 3D). Moreover, both DIP1-GFP protein and the fluorescence signal of DIP1-GFP were detected in the nucleus (Appendix Fig. S6E,F), supporting that DIP1 interacts with *DANA1* in the nucleus. These results together demonstrate that DIP1 can bind to *DANA1*.

## DIP1 is involved in plant drought response

To explore the role of DIP1 in plant responses to drought, we generated two independent *dip1* mutants, *dip1-1* and *dip1-2*, by using a CRISPR/Cas9-mediated gene editing (Appendix Fig. S7). The *DIP1* gene in the *dip1-1* line had both a 1-bp insertion and a 3-bp deletion in its target region, causing changed amino acids starting from the mutation site and resulting in premature

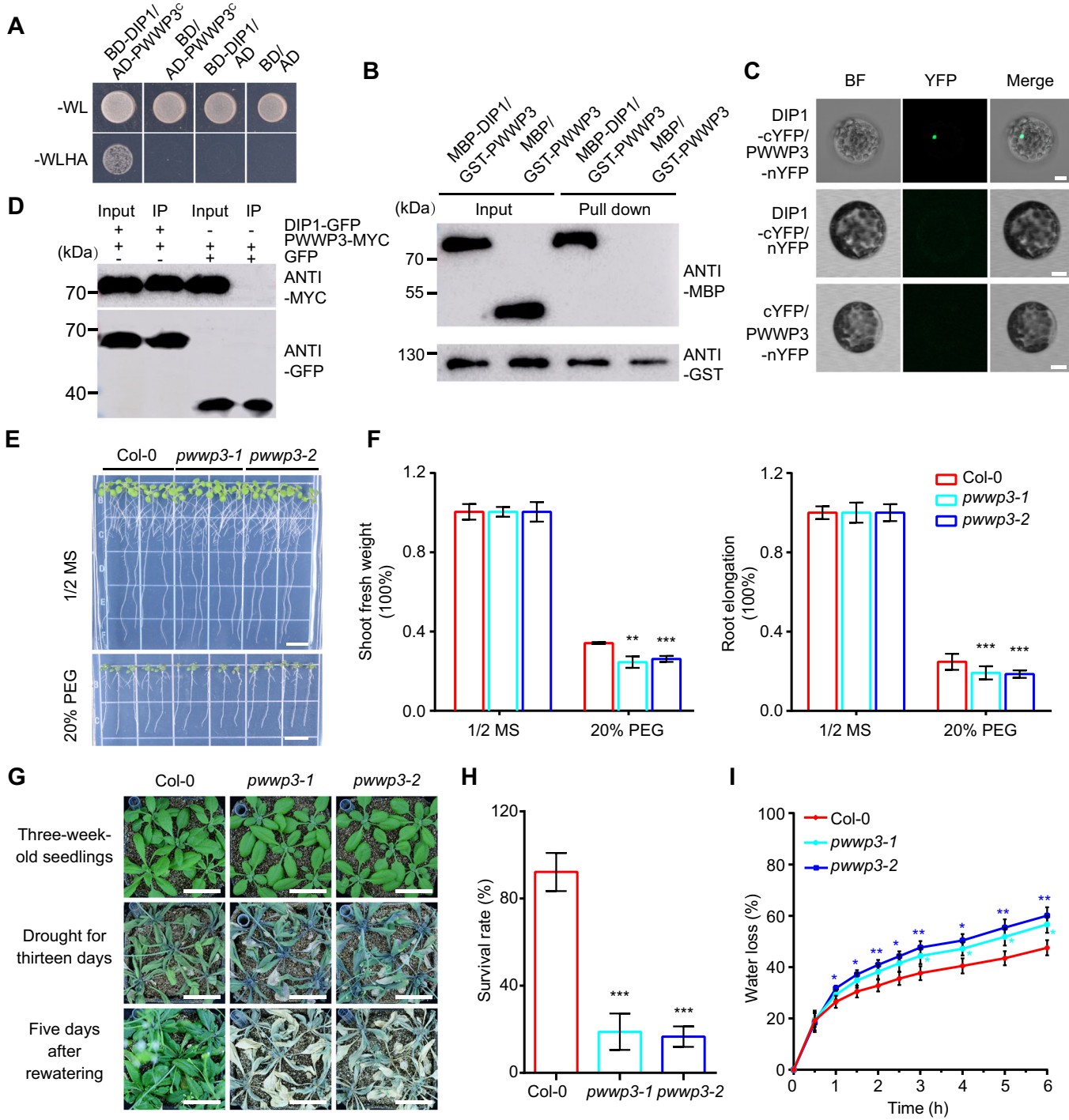

translation termination, whereas *dip1-2* had a 2-bp deletion in its target region, changing 4 altered amino acids downstream of the deletion site and truncating 66 amino acids in the protein (Appendix Fig. S7). Both *dip1-1* and *dip1-2* seedlings exhibited remarkably increased sensitivity when compared to WT under PEG treatment (Fig. 3E,F). Moreover, when the drought stress assay was carried out, two *dip1* mutant plants exhibited more drought sensitivity than Col-0 plants (Fig. 3G), which is consistent with the trend presented by *dana1* mutants. Compared with a ~93% survival

rate in the WT, only ~22% of *dip1-1* and ~17% of *dip1-2* plants survived after 2 weeks of drought stress followed by a 5-day recovery period (Fig. 3H). Water-loss analysis showed that the detached leaves of *dip1* mutants lost water more quickly than did those of Col-0 (Fig. 3I). In agreement, both stomatal opening and ABA-induced stomatal closure of *dip1-1* and *dip1-2* plants were strongly impaired when compared with that of the Col-0 plants (Appendix Fig. S8A,B). Moreover, we examined the expression of *CYP707A1*, *CYP707A2*, *CYP707A4*, *UGT71B1*, *MYB44* and *NTL6* in

**Figure 4. *PWWP3* mutants are more sensitive to drought stress.**

(A) DIP1 interaction with a C-terminal region of PWWP3 (aa 361–645) in yeast cells. (B) In vitro pull-down assay of MBP-DIP1 and GST-PWWP3. (C) BiFC (Bimolecular Fluorescence Complementation) assay of the interaction between DIP1 and PWWP3 in *Arabidopsis* protoplast cells. *Arabidopsis* mesophyll protoplasts were transformed with different plasmid pairs as follows. DIP1-cYFP/PWWP3-nYFP: co-transformation of DIP1 fused C terminus of YFP and PWWP3 fused N terminus of YFP; DIP1-cYFP/nYFP: co-transformation of DIP1 fused C terminus of YFP and N terminus of YFP; cYFP/PWWP3-nYFP: co-transformation of C terminus of YFP and PWWP3 fused N terminus of YFP. Scale bars = 10 μm. (D) Co-immunoprecipitation (Co-IP) assay of DIP1 and PWWP3. *PWWP3-MYC* was cotransformed with *DIP1-GFP* or *GFP* in *Nicotiana benthamiana* leaves. The expressed proteins were immunoprecipitated using an anti-GFP antibody and then detected with anti-MYC antibody. (E) *PWWP3* mutants are sensitive to PEG treatment. Scale bars = 1 cm. (F) Root length and fresh weight of seedlings shown in (E). Error bars represent standard deviation (*n* = 15, *n* refers to biological replicates). (G) Drought-tolerance assay. Col-0, *pwwp3-1* and *pwwp3-2* plants grown under normal growth conditions for 3 weeks were subjected to drought stress for 13 days and then rewatered for 5 days. Scale bars = 3 cm. (H) Survival rate after drought treatment (*n* = 3, *n* refers to biological replicates). (I) Water loss in detached leaves of 3-week-old Col-0, *pwwp3-1* and *pwwp3-2* plants. Data information: Values shown are means ± SD from three biological replicates. Asterisks represent significant differences determined by Student's *t* test (*$P < 0.05$; **$P < 0.01$; ***$P < 0.001$). Source data are available online for this figure.

*dip1* mutants, which showed the same behavior as in the *dana1* mutant background (Appendix Fig. S9A–F). Nevertheless, mutated *DIP1* did not influence the expression of *DANA1* (Appendix Fig. S9G).

In addition, PEG treatment and drought stress were performed on two *DIP1* overexpressing lines, designated *DIP1* OE-3 and *DIP1* OE-5 (Appendix Fig. S6A,B). We found that both *DIP1* OE-3 and *DIP1* OE-5 were more resistant to PEG and drought stress than the Col-0 (Appendix Fig. S6C,D,G,H), which is in agreement with the findings of *DANA1* overexpressing lines. Consistent with the drought-tolerant phenotype, detached leaves from *DIP1*-overexpressing plants showed reduced water loss (Appendix Fig. S6I). In *DIP1* overexpressing lines, the stomata aperture was clearly narrower when compared with that of the Col-0 plants (Appendix Fig. S8C,D). We then investigated the genetic interaction between *DANA1* and *DIP1* in the context of drought tolerance by generating *dana1-2*/*dip1-1* double mutant, *DANA1* overexpressing lines in *dip1-1* (*DANA1* OE-15 (*dip1-1*)) and *DIP1* overexpressing lines in *dana1-2* (*DIP1* OE-5 (*dana1-2*)). The *dana1-2*/*dip1-1* double mutant, *DANA1* OE-15 (*dip1-1*) and *DIP1* OE-5 (*dana1-2*) were drought-sensitive, consistent with the drought-sensitive phenotypes of both *dana1* and *dip1* mutants (Appendix Fig. S10A,B,D,E,G,H). Increased water loss was found in detached leaves of *dana1-2*/*dip1-1* double mutant, *DANA1* OE-15 (*dip1-1*) and *DIP1* OE-5 (*dana1-2*), which are in agreement with their drought-sensitive phenotypes (Appendix Fig. S10C,F,I). Taken together, these results indicate that *DANA1* works with DIP1 to regulate the plant drought response.

## Knockout of PWWP3, a protein interacting with DIP1, attenuates drought tolerance in *Arabidopsis*

To further characterize the molecular complex involving *DANA1* and DIP1 in response to drought, we performed a yeast two-hybrid assay to identify potential proteins interacting with DIP1. Strikingly, a C-terminal protein fragment of PWWP3 was identified as a DIP1-protein partner in yeast cells (Fig. 4A), and a subsequent protein pull-down experiment using GST-PWWP3 revealed that PWWP3 was able to pull down MBP-tagged DIP1 (MBP-DIP1), further validating their interaction in vitro (Fig. 4B). In addition, the interaction between DIP1 and PWWP3 was also confirmed in vivo by using a BiFC assay in *Arabidopsis* protoplast cells and a co-immunoprecipitation (Co-IP) assay in *Nicotiana benthamiana* (Fig. 4C,D). PWWP3 is part of the epigenetically active PEAT (PWWPs-EPCRs-ARIDs-TRBs) complex (Tan et al, 2018), consistent with the nuclear localization of the *DANA1*-DIP1 complex. Next, PEG treatment and drought stress assays were performed on

two T-DNA insertion mutant alleles, *pwwp3-1* and *pwwp3-2* (Tan et al, 2018). Phenotyping of *pwwp3-1* and *pwwp3-2* revealed that these two mutants were substantially more sensitive to PEG and drought stress when compared to WT plants (Fig. 4E–H), which is consistent with the observations from knockout of DIP1. In agreement with the drought-sensitive phenotype, detached leaves from *pwwp3* mutants exhibited increased water loss (Fig. 4I). Also, *pwwp3* mutants had a greater stomatal aperture index and enhanced ABA-induced stomatal closure when compared to WT (Appendix Fig. S11). These findings indicate that DIP1 regulates plant drought response together with PWWP3.

## *DANA1* regulates HDA9-mediated drought response

PWWP3 is known to belong to the PEAT complex that associates with HDA9 histone deacetylase (Tan et al, 2018), and HDA9 can regulate drought response through modulating expression of *CYP707A1* and *CYP707A2* in *Arabidopsis* (Baek et al, 2020). Therefore, the expression levels of *CYP707A1* and *CYP707A2* in *pwwp3* mutant were measured by RT-qPCR, and we found that both genes were upregulated in *pwwp3* mutant seedlings (Appendix Fig. S12). In agreement, we found that both H3K9ac and H3K27ac active marks were increased at the *CYP707A1* and *CYP707A2* loci in both *dana1-2* and *dip1-1* mutants (Fig. 5A–C). HDA9 is critical for the deacetylation of H3K9ac and H3K27ac, and our analysis of HDA9 binding around *CYP707A1* and *CYP707A2* loci revealed that the HDA9 enrichment on these chromatin regions is reduced in both *dana1* and *dip1* mutants (Fig. 5D; Appendix Fig. S13). Moreover, H3K9ac and H3K27ac profiles for both *CYP707A1* and *CYP707A2* loci in Col-0, *dana1-2*, and *dip1-1* plants under dehydration treatment were investigated, and we found that dehydration treatment causes the decreases of H3K9ac and H3K27ac levels and increase of HDA9 enrichment at both loci (Fig. 5A–D). To further determine the role of *DANA1* in HDA9-mediated drought stress, we carried out chromatin isolation by RNA purification (ChIRP) experiments. Two independent sets of biotinylated probes (EVEN and ODD) were used to purify *DANA1* RNA successfully, and one additional set of biotinylated probes (LacZ) was used as a negative control (Fig. 5E). By performing ChIRP-qPCR, we confirmed that *DANA1* RNA physically associates with DNA from the *CYP707A1* and *CYP707A2* loci, and dehydration treatment causes the increase of *DANA1* binding at both loci (Fig. 5F). Moreover, both interactions of *DANA1* RNA-*CYP707A1* DNA and *DANA1* RNA-*CYP707A2* DNA were reduced in *dip1-1* mutants (Appendix Fig. S14), indicating that DIP1 participates in *DANA1*-target recognition. Given that mutated DIP1 could impair

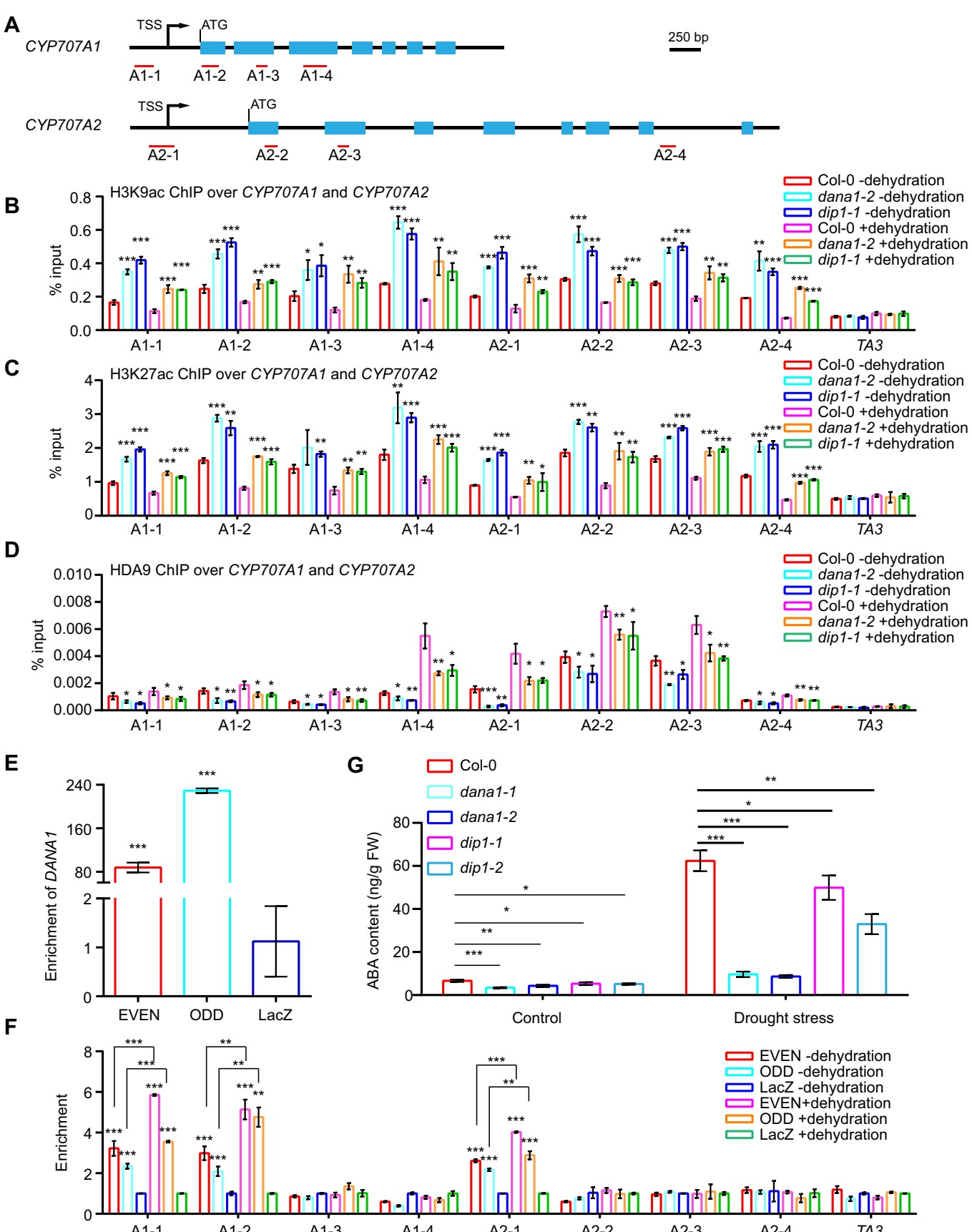

**Figure 5.  H3K9ac and H3K27 marks along the *CYP707A1* and *CYP707A2* loci are modulated by DANA1 regulation of HDA9 deposition.**

(A) Gene structure of *CYP707A1* and *CYP707A2*, indicating exons (boxes) and introns (lines). The locations of the gene regions analyzed by ChIRP-qPCR and ChIP-qPCR are marked. (B) The presence of H3K9ac was measured on *CYP707A1* and *CYP707A2* by ChIP-qPCR in WT, *dana1-2*, and *dip1-1* plants. (C) The presence of H3K27ac was measured on *CYP707A1* and *CYP707A2* by ChIP-qPCR in WT, *dana1-2*, and *dip1-1* plants. (D) The HDA9 enrichment at *CYP707A1* and *CYP707A2* chromatins depends on *DANA1* and DIP1. (E) Even and odd probes successfully retrieve *DANA1* RNA. LacZ probes are used in negative control. (F) *DANA1* association to DNA of *CYP707A1* and *CYP707A2* by ChIRP-qPCR in WT plants. Data from ChIRP-qPCR are represented relative to the background level of DNA precipitation (*PP2A*), and the *TA3* locus was used as the negative control (Li et al, 2016). (G) ABA content was measured in the rosette leaves of 3-week-old Col-0, *dana1* mutant, *dip1* mutant plants under 10 days of drought stress or not. (*n* = 3, *n* refers to biological replicates). Data information: Values shown are means ± SD from three biological replicates. Asterisks represent significant differences by Student's *t* test (*P < 0.05; **P < 0.01; ***P < 0. 001). Source data are available online for this figure.

the *DANA1* recognition of *CYP707A1* and *CYP707A2* (Appendix Fig. S14), the effect of *DANA1* on DIP1 binding to *CYP707A1* and *CYP707A2* has been investigated by ChIP-qPCR. We found that the DIP1 enrichment on *CYP707A1* and *CYP707A2* chromatin regions is reduced in the *dana1* mutant, and DIP1 did not bind to the *CYP707A4*, *UGT71B1*, *MYB44*, and *NTL6* loci (Appendix Fig. S15A–C). Therefore, our results hint at a cooperative binding of *DANA1* and DIP1 to common target genes. Collectively, these findings show that *DANA1* and DIP1 together regulate the enrichment of HDA9 across *CYP707A1* and *CYP707A2* loci.

## Discussion

Drought stress is a major threat to crop production, and plants respond to drought stress by complex adjustments, including stomatal closure and induction of drought-responsive genes. Besides protein-coding genes and miRNAs, a large number of lncRNAs exhibited drought-responsive expression patterns in *Arabidopsis* (Liu et al, 2012) and rice (Chung et al, 2016). However, studies regarding the role of lncRNAs involved in drought response are very limited. Changing the expression of *StFLORE* which is a natural antisense transcript of *StCDF1* had powerful effects on water homeostasis in potato, and this response is due to the regulation of stomatal opening in an ABA-dependent manner (Ramírez Gonzales et al, 2021). *Cis-NATZmNAC48* could negatively regulate *ZmNAC48* expression at the post-transcriptional level, and nat-siRNAs from the overlapping region of *cis-NATZmNAC48* and *ZmNAC48* transcripts were identified by sequencing small RNAs (Mao et al, 2021). In *Arabidopsis thaliana*, increasing the lncRNA *DRIR* expression enhanced plant drought tolerance, and transcriptomic analyses showed that *DRIR* modulates the expression of genes involved in the stress response, including ABA response, water transport and transcription (Qin et al, 2017). Nonetheless, the underlying mechanisms of lncRNA-mediated plant drought response remain poorly understood. In this study, several lines of evidence show that the lncRNA *DANA1* is involved in plant responses to drought by binding to the novel epigenetic regulator DIP1 and affecting HDA9 binding to stress-responsive loci. Interestingly, another cytosol-localized lncRNA encoded by AT4G14548 locates on upstream of *DANA1* (Engelmann, 2022), and we investigated its association with *DANA1*-meiated drought response. Expression level of AT4G14548 gene in *dana1-2* was measured by RT-qPCR, and we found that AT4G14548 expression was significantly upregulated in *dana1-2* mutant (Appendix Fig. S16). Then we examined AT4G14548 expression levels in *DANA1* COM plants, and found that its transcriptional levels were even higher than in *dana1-2* mutant

(Appendix Fig. S16). However, expression levels of *CYP707A1* and *CYP707A2* in *DANA1* COM were similar to Col-0 wild-type (Appendix Fig. S16). Because *DANA1* COM showed the same phenotype as the Col-0 wild-type according to the drought-tolerance assay (Appendix Fig. S3D–F), these results together indicate that AT4G14548 is not involved in *DANA1*-mediated drought response in *Arabidopsis*. Our findings are also supported by Engelmann's result that AT4G14548 was still expressed in plants with a 1538-bp deletion, in which whole *DANA1* genomic sequence are removed (Engelmann, 2022). For upregulation of AT4G14548 in *dana1* mutants, this is might be caused by T-DNA insertion. For upregulation of AT4G14548 in *DANA1* COM, we suspect that it is caused by both T-DNA insertion and insertional position of *DANA1* complemented construct. Based on our results, we propose the following model for the *DANA1*-mediated drought response in plants. When plants are grown under normal conditions, *CYP707A1* and *CYP707A2* containing high levels of H3K9ac and H3K27ac are highly expressed (Fig. 6). However, in the presence of drought stress, the transcription of *DANA1* is upregulated and its transcripts bind to DIP1 in the nucleus, resulting in an increase of HDA9 bound to *CYP707A1* and *CYP707A2* loci by means of PWWP3 (Fig. 6). By increasing the binding of HDA9 to *CYP707A1* and *CYP707A2* loci, HDA9 diminishes H3K9ac and H3K27ac on both loci, leading to the transcriptional downregulation of *CYP707A1* and *CYP707A2* (Fig. 6). Our model proposes that *DANA1* can regulate *CYP707A1* and *CYP707A2* expression during the plant response to drought by regulating histone deacetylation.

Under drought stress, plants accumulate ABA. Indeed, we observed that ABA content in *dana1* mutant plants was markedly reduced compared to WT under normal and drought stress conditions, and the reduced ABA content was also observed in *dip1* mutant plants (Fig. 5G). The ABA content in *DANA1* OE plants was significantly higher than in Col-0 WT (Appendix Fig. S17). Both stomatal index and stomatal density were significantly decreased in *DANA1* OE plants compared with WT (Appendix Fig. S18), which is consistent with the phenotype of *cyp707a1/cyp707a3* double mutant (Tanaka et al, 2013). It has been known that CYP707A2 plays an essential role in seed dormancy and germination (Kushiro et al, 2004), thus, we tested whether *DANA1* plays a role in seed germination. Mutants of *dana1* exhibited insensitivity to PEG treatment at the stage of germination (Appendix Fig. S19). Moreover, expressional levels of *DANA1* during pre- and post-germination stages in Col-0 were examined. *DANA1* mRNA was downregulated immediately after imbibition, and decreasing gradually (Appendix Fig. S20A). However, in the plant post-germination stage, the expression of *DANA1* gradually increased (Appendix Fig. S20B). Then effects of *DANA1* on stomatal aperture index, stomatal conductance, transpiration rate,

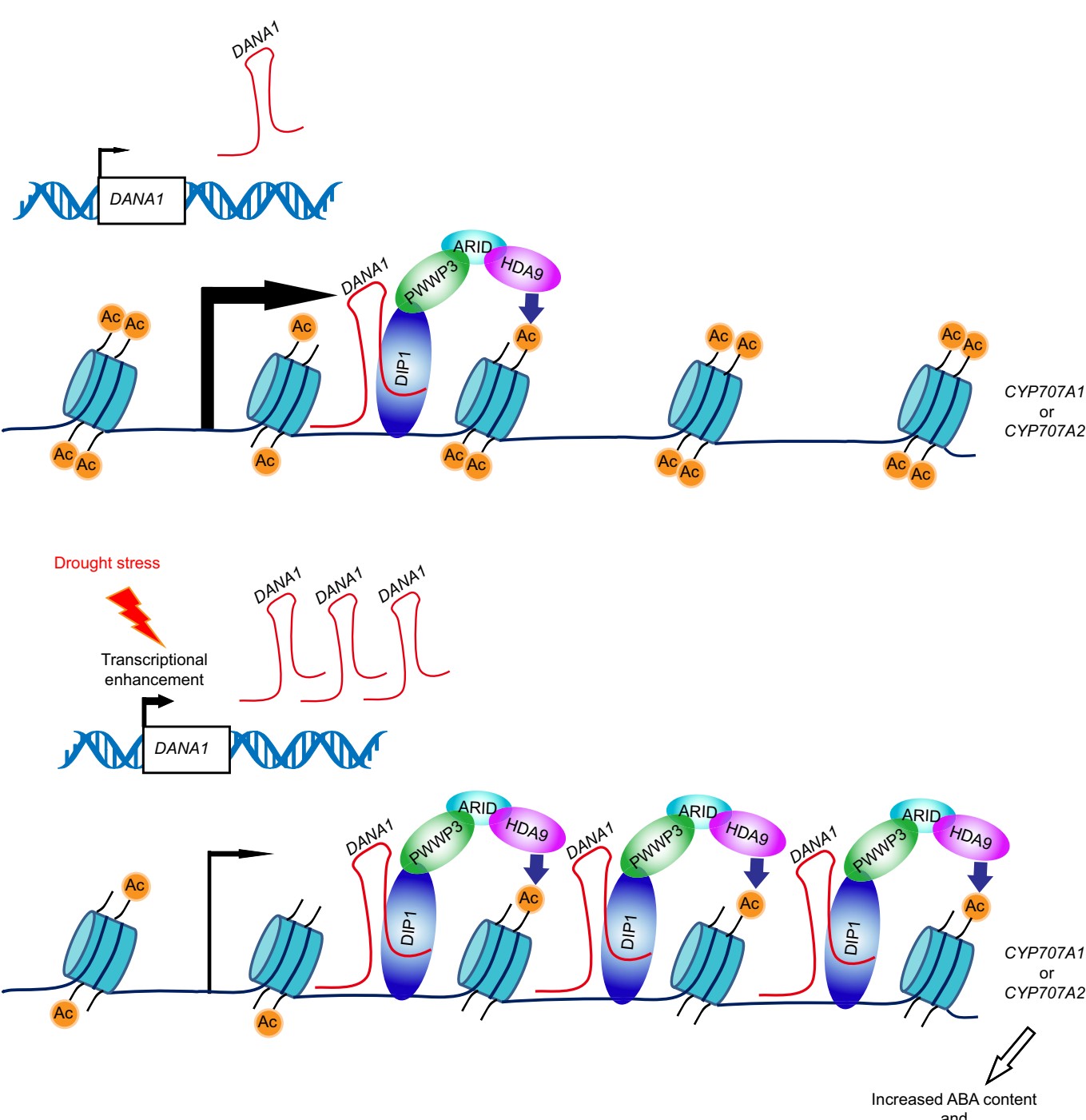

**Figure 6. A proposed working model of *DANA1* function in plant responses to drought.**

Under normal conditions, *CYP707A1* and *CYP707A2* with high levels of H3K9ac and H3K27ac modifications are highly expressed. In the presence of drought stress, the *DANA1* transcript level increases and binds to DIP1, leading to the enrichment of HDA9 at *CYP707A1* and *CYP707A2* loci through interaction with a PEAT complex component, PWWP3. Consequently, H3K9ac and H3K27ac levels at both loci are reduced, and *CYP707A1* and *CYP707A2* expressions are repressed, causing the ABA content to increase and enhancement of drought tolerance in plants.

and water use efficiency in *Arabidopsis* plants before and after drought treatment were studied. We found that stomatal aperture index, stomatal conductance and transpiration rate were significantly increased in *dana1* mutants, but water use efficiency was significantly decreased in *dana1* mutants (Appendix Fig. S21). These results further indicate that *DANA1* regulates plant drought response by modulating the expression of ABA catabolic genes including *CYP707A1* and *CYP707A2*. Given that ABA-dependent stress responses largely depend on endogenous ABA levels in the plant, which fluctuates widely in response to drought stress (Seki et al, 2007), we employed RT-qPCR to measure *DANA1* RNA abundance in the ABA signaling mutant, *pyr1/pyl1/pyl4* triple mutant. Expression of *DANA1* in *pyr1/pyl1/pyl4* plants was significantly higher than in Col-0 plants under both control conditions and dehydration treatment (Appendix Fig. S22), indicating that *DANA1* expression might be negatively regulated by ABA signaling. The connection between *DANA1*-mediated drought response and ABA signaling pathway shall be explored in future. In addition, *DANA1* did not directly interact with PWWP3, ARIDs and HDA9 in yeast cells (Appendix Fig. S23A), and the in vitro pull-down assay showed that *DANA1* has no effect on the interaction between DIP1 and PWWP3 (Appendix Fig. S23B). Then, the interaction between DIP1 and HDA9 was investigated by yeast two-hybrid assay, and we found that DIP1 could not directly interact with HDA9 in yeast cells (Appendix Fig. S24). Considering the drought-sensitive phenotypes exhibited by *dana1*, *dip1*, and *pwwp3* mutants, these three genes may act on the same checkpoint in drought response, and further genetic analysis of genetic interaction among them will be needed in the future.

RNA-binding proteins (RBPs) play important roles in a wide range of biological processes, a large number of canonical RBPs and RBPs lacking conventional RNA-binding regions (RBRs) have been uncovered by recent proteome-wide studies, but their biological roles have not been fully characterized. In our study, we have identified a novel RNA-binding protein, DIP1, which was previously annotated as a ribosomal protein belonging to the L1p/L10e family. Ribosomal proteins serve a largely structural role in ribosomes, and exert an important function in the cellular process of translation. However, our findings prove that lncRNAs can interact with L1p/L10e-type proteins to regulate gene expression in the nucleus by altering histone deacetylation. This demonstrates a diversification in subcellular localization and expands the range of roles of L1p/L10e proteins, notably identifying them as novel epigenetic regulators of gene expression. Phylogenetic analysis also shows that DIP1 has diverged from other members of the L1p/L10e family that are components of the 80 S ribosome (Appendix Fig. S25), indicating that DIP1 is not a canonical ribosome component. In *Arabidopsis*, the lncRNA *APOLO* interacts with the plant PRC1 component LHP1 in the regulation of a subset of common genes in trans across the *Arabidopsis* genome. It was shown that *APOLO* modulates LHP1 binding to chromatin, although *lhp1* knockout does not affect the R-loop mediated interaction of *APOLO* with its target loci (Ariel et al, 2020). Interestingly, it was recently demonstrated that *APOLO* also recruits the transcription factor WRKY42 to common targets involved in root hair development (Moison et al, 2021), whereas *wrky42* mutation does not affect *APOLO* binding to DNA either. Here we demonstrate that in the *dip1* background, *DANA1* recognition of target genes is impaired, suggesting that DIP1 is a novel actor in lncRNA recruitment to common target genes. Further research will be required to uncover the

general role of L1p/L10e proteins in chromatin dynamics and RNA-mediated mechanisms regulating gene expression. In addition, RNA secondary structure has been reported to be associated with the functional role for lncRNA (Hawkes et al, 2016; Wang et al, 2014). We used the *RNAfold* service to analyze the RNA secondary structure of *DANA1*, and then divided it into three fragments (*DANA1*-L1, *DANA1*-L2, and *DANA1*-L3) depending on its predicted structure (Appendix Fig. S26A). Then a yeast three-hybrid assay was performed on the full-length *DANA1* and its three fragments, and we found that both the full-length *DANA1* and *DANA1*-L1 can interact with DIP1 in yeast cells (Appendix Fig. S26B), suggesting that the RNA fragment between nucleotides 1 and 175 of *DANA1* is important for the interaction between *DANA1* and DIP1. Moreover, with recent advances in technologies for profiling RNA structure and RBP binding sites (Gosai et al, 2015; Yang et al, 2018), it will be informative to directly identify the DIP1 binding sites in *DANA1* and the secondary structure of *DANA1* to explore the detailed interacting mechanism between DIP1 and *DANA1* in the future. In addition, considering that genes involved in "Defense response" were significantly overrepresented based on GO enrichment analysis of the DEGs (see Dataset EV2), therefore, it will be interesting to study the role of *DANA1* in plant responses to pathogens henceforth.

In the Celtic mythology, Dana (or Danu) is considered the goddess of fertility and is associated with water and rivers. Here we have characterized the *DANA1*-regulated drought response in *Arabidopsis* and uncovered a new lncRNA-integrating complex, including DIP1 and PWWP3, which modulates gene expression by influencing histone deacetylation. We have demonstrated that lncRNAs can play an important role in the epigenetic reprograming of plant responses to stress.

# Methods

## Plant materials and growth conditions

*Arabidopsis* plants used in this study were under the Col-0 background. The *dana1-1* (CS833797), *dana1-2* (SALK_146574) and *hda9-1* (SALK_007123) were obtained from *Arabidopsis* Biological Resource Center (ABRC), and T-DNA insertion lines including *pwwp3-1* (SALK_042581) and *pwwp3-2* (CS836957) were kindly provided by Prof. Xin-Jian He (Tan et al, 2018). For CRISPR/Cas9-induced mutants, sgRNAs were designed by applying the Cas-Designer (http://www.rgenome.net/cas-designer/). For producing overexpressing lines, the full-length region of *DANA1* and the full-length of *DIP1* coding sequence were cloned into the vectors pCambia1300-S1 and pCambia1300-GFP-HA, respectively. For generating *DANA1* COM, DNA fragments including 2 kb upstream of *DANA1* and the full-length *DANA1* sequence plus a 300-bp downstream sequence were amplified from Col-0 genomic DNA, and then cloned into the vectors pCambia1300. All produced plasmids were introduced into the *Agrobacterium tumefaciens* strain GV3101, and then transformed into *Arabidopsis thaliana* plants *via* the flower dipping method. The *dana1-2* was applied for producing *DANA1* COM. For *DIP1* OE (*dana1-2*), it is obtained by crossing *DIP1* OE-5 with *dana1-2*. For *dana1-2/dip1-1*, it is obtained by crossing *dana1-2* with *dip1-1*. For *DANA1* OE (*dip1-1*), it is obtained by crossing *DANA1* OE-15 with *dip1-1*. Primers used for vector and sgRNA constructions are listed in Appendix Table S1.

*Arabidopsis* seeds were surface-sterilized with 70% ethanol for 10 min, and washed with sterile water for five times. Then, seeds were sown on half MS media, stratified for 3 days at 4 °C in the dark, and grown at 22 °C with a 16-h light/8-h dark photoperiod.

## Stress treatment

PEG assay was performed according to an earlier study (Wang et al, 2011). The PEG-infused plates were prepared as described by Verslues et al (2006). An overlay solution containing PEG was poured over 1/2 MS agar plates, and PEG was allowed to diffuse into the medium. Four-day-old seedlings grown on half MS media were transferred onto 1/2 MS media supplemented with or without 20% (w/v) PEG, and pictures were taken after 6 days.

For the drought-tolerance test, ~3-week-old well-watered seedlings had watering withheld for the number of indicated days, and then were rewatered to allow recovery for 5 days. In addition, the same amount of soil was applied in each pot, and pot positions were changed randomly every 2 days. The plants before and after the treatment were photographed and surveyed according to the previous description (Wang et al, 2011). For measurement of water loss in detached leaves, fully expanded rosette leaves were removed from 3-week-old well-watered plants and placed at the same conditions used for seedling growth, and each sample (consisting of five individual leaves) was weighed at the indicated times as previously described (Wang et al, 2011).

## Stomatal aperture, stomatal conductance, transpiration rate, water use efficiency, stomatal index, and stomatal density measurement

Stomatal aperture measurement was conducted as described previously with small modifications (Eisele et al, 2016). In brief, for ABA-induced stomatal closure assays, detached rosette leaves of 3-week-old plants were incubated in a stomatal open solution (5 mM KCl, 50 μM CaCl$_2$, 10 mM MES-KOH, pH 6.15) for 3 h, and then these leaves were transferred to the stomatal open solution supplemented with 50 μM ABA and incubated under light for 0, 30, or 60 min. The detached leaves of 3-week-old plants with or without treatment for inducing stomatal closure were dipped into the staining solution (1 μM rhodamine 6 G, 4% formaldehyde) for 10 min and photographed under a Nikon ECLIPSE Ti2-U microscope. The width and the length of the stomatal aperture were measured with ImageJ software, and the stomatal aperture index was calculated with the following formula: stomatal aperture index = width of aperture/length of aperture.

Instantaneous measurements of stomatal conductance, transpiration rate and water use efficiency were performed by applying the LI-6800XT Portable Photosynthesis System (*LI-CORE*, USA). Leaves were placed in the chamber with an ambient CO$_2$ concentration of 400 μmol mol$^{-1}$ and a temperature of 22 °C, and irradiance was set to 150 mmol m$^{-2}$ s$^{-1}$. A reading was recorded when the IRGA (infrared gas analyzer) conditions had stabilized. Intrinsic water use efficiency (WUEi) was calculated with the following formula: WUEi = Photosynthesis rate/transpiration rate.

Measurements of the stomatal index and stomatal density were performed as described previously (Xiao et al, 2021). In brief, the cotyledons of 5-day-old seedlings were analyzed for stomatal index and stomatal density, and the central areas derived from the abaxial

cotyledon surface were imaged using an LSM 900 confocal laser scanning microscope (Zeiss). Stomata and pavement cell numbers were counted with ImageJ software, and the stomatal index was calculated by dividing the number of stomata by the total amount of cells (stomata + pavement cells).

## ABA content measurement

For quantification of ABA content, rosette leaves from 3-week-old seedings of Col-0, *dana1* mutant and *dip1* mutant grown under normal and 10 days of drought stress conditions, and 10-day-old seedlings of Col-0 and *DANA1* overexpressing lines grown on 1/2 MS agar plates were collected, then sent to the SanShu Biotech company (Shanghai, China) for UHPLC-ESI-MS/MS analysis. The standard sample used was ABA (Sigma), and each sample was measured with three biological replicates.

## Differential gene expression analysis

Differential gene expression analysis was carried out using the edgeR (Robinson et al, 2010). A threshold of *P* value < 0.05 was used to select statistically significant differentially expressed genes. Gene ontology enrichment analysis was performed using a web tool agriGO v2.0 with the following settings: statistical test method as "Hypergeometric", multi_test adjustment method as "Hochberg FDR", significance level as "0.05", minimum number of mapping entries as "5" and gene ontology type as "Complete GO" (Tian et al, 2017). The circos plot of differentially expressed genes was generated using a R package RCircos (Zhang et al, 2013).

## Yeast three-hybrid assay and yeast two-hybrid assay

Yeast three-hybrid assay with *DANA1* was performed as previously described with small modifications (SenGupta et al, 1996). The full-length DNAs of *DANA1* and *TE-lincRNA11195* were introduced into the vector pIII/MS2-1, respectively. After that the resultant pIII/MS2-*DANA1* and pIII/MS2-*TE-lincRNA11195* were separately transformed into the yeast strain YBZ1. The yeast strains were then transformed with yeast three-hybrid RNA-binding protein cDNA library constructed by our laboratory, and transformants were selected on media lacking Leu, Ura and His. For examining the interaction between *DANA1* and PWWP3, HDA9, ARID2, ARID3 and ARID4, the full-length coding sequences of PWWP3, HDA9, ARID2, ARID3 and ARID4 were amplified with RT-PCR. Those DNA sequences were separately cloned into pGADT7, and transferred into the YBZ1 for the following yeast three-hybrid assay.

For yeast two-hybrid assay, the full-length *DIP1* coding sequence was introduced into the vector pGBKT7, and then transferred into the yeast strain AH109. The yeast strains were transformed with the normalized Arabidopsis cDNA library (Mate & Plate Library, Universal Arabidopsis, Clontech), and transformants were selected on media lacking Leu, Trp, His, and Ade.

## In vitro RNA pull-down assay and RIP assay

The tRSA RNA pull-down assay was carried out as described previously with some modifications (Iioka et al, 2011). Both tRSA and tRSA-*DANA1* were in vitro transcribed using the TranscriptAid T7

High Yield Transcription Kit (Thermo Scientific), and then purified with the GeneJET RNA purification Kit (Thermo Scientific). The pull-down assay was performed applying Pierce Magnetic RNA-Protein Pull-Down Kit (Thermo Scientific) with 50 picomole (pM) of RNA and 5 μg of soluble protein. Retrieved proteins were detected by western blotting with primary antibody anti-GST (Abmart).

RIP assays with DIP1-GFP lines were conducted as previously described (Zhao et al, 2018). Briefly, 10-day-old seedlings were harvested and cross-linked by using 1% formaldehyde for 10 min. RNA-protein complexes were immunoprecipitated with antibodies mouse IgG (Cell Signaling Technology) and anti-GFP (Roche) at 4 °C for 1 h on a rotation mixer. Dynabeads Protein G (Invitrogen) was applied to capture the immunocomplexes. At last, the cross-linking was reversed and RNA was extracted by Trizol (Ambion). Primers used for RNA pulldown and RIP are listed in Appendix Table S1.

## TriFC assay and BiFC assay

Binary TriFC vectors were produced according to an earlier study for transient expression in tobacco (Seo et al, 2017). DIP1, MSCP and DANA1 were cloned into pCambia1301-nYFP, pCambia1301-cYFP and pCambia1301-6xMS2, respectively. All plasmids were transformed into Agrobacterium strain EHA105 using the freeze-and-thaw method. Cultured cells were harvested and re-suspended in the infiltration buffer (10 mM $MgCl_2$ and 200 mM acetosyringone), and then kept at room temperature for more than 3 h. Agrobacterium suspensions, including 50 μM MG132 were infiltrated into 3-week-old leaves of N. benthamiana, and leaf cells were analyzed using an LSM 900 confocal laser scanning microscope (Zeiss) at 2 days after infiltration.

The BiFC assay was performed as previously described (Seo et al, 2017). Concisely, DIP1 and PWWP3 were cloned into the pVYCE (cYFP) vector and pVYNE (nYFP) vector, respectively. All constructs were transformed into Arabidopsis protoplasts by PEG-mediated transfection (Yoo et al, 2007), and then transfected cells were examined using an LSM 900 confocal laser scanning microscope (Zeiss).

## Nuclear-cytoplasmic fractionation

Ten-day-old seedlings of Col-0 were used for nuclear-cytoplasmic fractionation according to the previous description (Tang et al, 2021). As quality controls for the fractionation, glutamine tRNA and U6 RNA were used as cytoplasmic marker and nuclear marker, respectively.

## CRISPR/Cas9-mediated mutagenesis

The constructs for CRISPR were designed according to the protocol described previously (Zhang et al, 2016). The sgRNAs for CRISPR/Cas9 vectors to edit DIP1 are listed in Appendix Table S1. The plasmid was transformed into Agrobacterium tumefaciens strain GV3101, and then transformed into A. thaliana Col-0 plants via the floral dip method.

## In vitro protein pull-down assay

In total, 500 μg GST-PWWP3 and 500 μg MBP or MBP-DIP1 were mixed into 1 ml of TGH buffer (50 mM HEPES pH 7.5, 150 mM NaCl, 1.5 mM $MgCl_2$, 1 mM EGTA pH 7.5, 1% Triton, 5% glycerol, 1 mM PMSF and protease inhibitor cocktail (Roche)) and incubated at 4 °C for 1 h. After incubation, reaction mixtures were pulled down with 120 μl of Glutathione Sepharose 4B (GE) for 1 h, then washed five times with the TGH buffer. The pulled-down proteins were analyzed on 10% SDS-PAGE gels, and then subjected to immunoblotting with anti-MBP (Abmart) or anti-GST (Abmart) antibodies.

## Co-IP assay

Co-IP assay was conducted as previously described (Yang et al, 2023). Protein extracts from infiltrated leaves were immunoprecipitated with anti-GFP (Roche). After washing the beads, the immunoprecipitated proteins were resolved by SDS-PAGE and detected by immunoblotting with anti-GFP (BBI life sciences) or anti-MYC (Abmart). For Co-IP assay performed with infiltrated leaves, agrobacteria harboring UBQ10: DIP1-GFP, UBQ10: GFP, and 35 S: PWWP3-MYC constructs described above were transiently expressed in leaves of N. benthamiana.

## RT-qPCR and western blotting

Total RNA was extracted using Trizol (Ambion), and the first-stand complementary DNA was synthesized by TransScript One-Step gDNA Removal and cDNA Synthesis SuperMix (Transgen). RT-qPCR was performed using SYBR qPCR Master Mix (Vazyme) on a CFX96 Real-Time PCR Detection System (Bio-Rad), and RT-qPCR primers are available in Appendix Table S1.

Total protein extracts were loaded on the 12% SDS-PAGE gels for protein separation. After transfer to nitrocellulose (GE Healthcare) membranes, proteins were detected using antibodies anti-ACTIN (BBI life sciences) and anti-GFP (BBI life sciences).

## Chromatin immunoprecipitation

Approximately 1.5 g of 10-day-old seedlings were collected without cross-linking and stored at −80 °C until use. ChIP was executed according to the previous study (Qiu et al, 2019), and the anti-HDA9 antibody was produced by ABclonal Biotech. The sonicated chromatin extractions were immunoprecipitated with antibodies anti-H3K9ac (Abcam), anti-H3K27ac (Abcam), and anti-HDA9 for Col-0, dana1-2 and dip1-1 plants at 4 °C for 4 h on a rotation mixer. For DIP1 OE (dana1-2) plants, antibodies mouse IgG (Cell Signaling Technology) and anti-GFP (Roche) were applied. Dynabeads Protein A (Invitrogen) was applied to capture the immunocomplexes with antibodies anti-H3K9ac and anti-H3K27ac, and Dynabeads Protein G (Invitrogen) was applied to capture the immunocomplexes with antibodies anti-HDA9, mouse IgG and anti-GFP. After reverse cross-linking and proteinase K (Merck Millipore) digestion, DNAs were extracted with phenol-chloroform and the precipitated with ethanol. Primers used for ChIP-qPCR are listed in Appendix Table S1.

## Chromatin isolation by RNA purification followed by qPCR

The ChIRP assay was carried out as described previously (Ariel et al, 2014), and antisense biotinylated DNA probes were designed

against *DANA1* full-length sequence applying an online designer at http://singlemoleculefish.com/. Sixteen probes were produced and split into two sets based on their relative positions along *DANA1* sequence, such as odd-numbered and even-numbered probes were separately pooled. Asymmetrical set of biotinylated DNA probes against LacZ RNA was produced as the mock control (Chu et al, 2011). In brief, 4 g of 10-day-old Col-0 seedlings were in vivo cross-linked and cell nuclei were purified and extracted through sonication. The resulting supernatant was hybridized against three set of biotinylated DNA probes separately, and then isolated by using Dynabeads™ MyOne™ Streptavidin C1 (Invitrogen). Co-purified ribonucleopreotein complexes were eluted and used to extract RNA or DNA, which were later subjected to the following experiments for quantification. Probes and primer used in ChIRP-qPCR are provided in Appendix Table S1.

## Data availability

RNA-seq data: NCBI's GEO database repository GSE186546.

## Peer review information

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

## Acknowledgements

We would be very grateful to Professor Xin-Jian He (National Institute of Biological Sciences, Beijing, China) for generously providing the seeds of *pwwp3-1* and *pwwp3-2*, and thank Prof. Jian-Kang Zhu for advising with the drought-tolerance test. This study was supported by the National Natural Science Foundation of China (32060067, 31800224, 31960138, and 32070627) and the Natural Science Foundation of Jiangxi Province (20171ACB20001).

## Author contributions

**Jingjing Cai**: Resources; Formal analysis; Validation; Investigation; Visualization; Methodology; Writing—original draft; Writing—review and editing. **Yongdi Zhang**: Resources; Formal analysis; Validation; Investigation; Methodology. **Reqing He**: Formal analysis; Visualization; Methodology; Writing—original draft; Writing—review and editing. **Liyun Jiang**: Formal analysis; Visualization; Methodology. **Zhipeng Qu**: Data curation; Writing—review and editing. **Jinbao Gu**: Validation; Investigation. **Jun Yang**: Validation; Investigation. **María Florencia Legascue**: Validation; Investigation. **Zhen-Yu Wang**: Writing—review and editing. **Federico Ariel**: Writing—review and editing. **David L Adelson**: Writing—review and editing. **Youlin Zhu**: Writing—review and editing. **Dong Wang**: Conceptualization; Formal analysis; Supervision; Methodology; Writing—original draft; Project administration; Writing—review and editing.

## Disclosure and competing interests statement

The authors declare no competing interests.

