## [Peer Review File · EMBO Reports]

LncRNA DANA1 promotes drought tolerance and histone deacetylation of drought responsive genes in Arabidopsis

Jingjing Cai, Yongdi Zhang, Reqing He, Liyun Jiang, Zhi Peng Qu, Jinbao Gu, Jun Yang, María Florencia Legascue, Zhen-Yu Wang, Federico Ariel, David Adelson, Youlin Zhu, and Dong Wang

DOI: [10.15252/embr.202357737](https://doi.org/10.15252/embr.202357737)

Corresponding author(s): Dong Wang (dongwang@ncu.edu.cn)

Review Timeline:

Submission Date:	29th Jun 23
Editorial Decision:	17th Aug 23
Revision Received:	10th Nov 23
Editorial Decision:	28th Nov 23
Revision Received:	29th Nov 23
Accepted:	4th Dec 23

Editor: Esther Schnapp

Transaction Report:

Dear Dr. Wang,

Thank you for your patience while your manuscript was peer-reviewed at EMBO reports. We have now received the full set of referee reports as well as referee cross-comments that are all pasted below.

As you will see, the referees acknowledge that the findings are potentially interesting. However, they also have several suggestions for how the data could be strengthened. It becomes clear from the cross-comments, that not all requests have to be addressed, for example, the DANA1-ChIPseq is not required. Also some of referee 3's comments do not have to be addressed experimentally. Referee 2 notes that the first point by referee 3 could be addressed, and referee 1 feels that you should address referee 3's points experimentally to the best of your abilities. If you like, we can also discuss the revisions in a video chat.

I would thus like to invite you to revise your manuscript with the understanding that the referee concerns must be fully addressed and their suggestions taken on board. Please address all referee concerns in a complete point-by-point response. Acceptance of the manuscript will depend on a positive outcome of a second round of review. It is EMBO reports policy to allow a single round of major revision only and acceptance or rejection of the manuscript will therefore depend on the completeness of your responses included in the next, final version of the manuscript.

We realize that it is difficult to revise to a specific deadline. In the interest of protecting the conceptual advance provided by the work, we recommend a revision within 3 months (17th Nov 2023). Please discuss the revision progress ahead of this time with the editor if you require more time to complete the revisions.

- 1) A data availability section providing access to data deposited in public databases is missing. If you have not deposited any data, please add a sentence to the data availability section that explains that.
- 2) Your manuscript contains statistics and error bars based on $n=2$. Please use scatter blots in these cases. No statistics should be calculated if $n=2$.

3) We replaced Supplementary Information with Expanded View (EV) Figures and Tables that are collapsible/expandable online. A maximum of 5 EV Figures can be typeset. EV Figures should be cited as 'Figure EV1, Figure EV2' etc... in the text and their respective legends should be included in the main text after the legends of regular figures.

5) a complete author checklist, which you can download from our author guidelines <https://www.embopress.org/page/journal/14693178/authorguide>. Please insert information in the checklist that is also reflected in the manuscript. The completed author checklist will also be part of the RPF.

6) Please note that all corresponding authors are required to supply an ORCID ID for their name upon submission of a revised manuscript (<https://orcid.org/>). Please find instructions on how to link your ORCID ID to your account in our manuscript tracking system in our Author guidelines <https://www.embopress.org/page/journal/14693178/authorguide#authorshipguidelines>

I look forward to seeing a revised form of your manuscript when it is ready.

Yours sincerely,

Esther Schnapp, PhD

Referee #1:

The manuscript titled 'LncRNA DANA1 promotes histone deacetylation of drought responsive genes in Arabidopsis' by Cai, Zhang, He and Jiang et al describes the role of long non coding RNA DANA1 in regulating drought stress response. They showed that it interacts with DIP1 which is part of a protein complex with PWWP3 and HDA9 which eventually regulates the histone enrichment of drought responsive CYP707A1 and CYP707A2 loci.

Though the manuscript highlights the emerging role of lncRNAs in drought stress, some key issues need to be addressed.

Major points:

1. It is interesting to study the *dana1* mutant as plants with enhanced ABA catabolism. It is obvious (Fig. 5G) that *dana1* mutants have reduced ABA levels under control conditions which increase in response to drought, albeit to a lower level. So, it is worthwhile to discuss that *dana1* may not be having a drought specific phenotype only but a general ABA deficient phenotype. Is drought tolerance just one of many responses as *dana1* mutants might be highly pleiotropic as seen in many epigenetic machinery mutants. In this regard, it will be great to study the DANA1 expression during different plant developmental (pre- and post-germination) stages. *dana1* mutants can be tested in detail for their germination, stomatal development and transpiration rate and water use efficiency during seedling and adult plant stages before and after PEG or drought treatment.

2. The good test that DANA1 regulates the CYP707 expression, would be to test the ABA levels in DANA1 OE lines. In that respect it will be interesting to check whether OE lines have a *cyp707* mutant like phenotype like Decreased stomatal index and density (Tanaka et al Plant J 2013).

3. Are the drought phenotype in *dana1* and *dip1* mutants ABA dependent? Can authors comment on what regulates the expression of DANA1 itself during drought treatment. Is there a possibility of a feedback loop regulation. It will be great to test the expression of DANA1 in ABA signaling mutants like *pyl/pyr*.

4. Though authors have nicely tested the histone enrichment in Fig.5, however, it will be great to test the dynamic histone acetylation before and after the drought treatment around these loci.

5. As DIP is a ribosome machinery interacting protein, does the authors check the effect of DIP mutation on translation of the proteins. Can the drought phenotype in *dip* mutant be partly because of translation impairments in the mutant. As the ribosomal machinery is localized in the cytoplasm, does the authors check the localization of DIP alone.

Minor Points:

1. The authors should introduce and discuss the ABA catabolism and CYP707 mutants phenotypes in detail.

2. Please explain in detail how the PEG plates were made in Methods section.

3. Why authors choose 10 d time point for ChIPs while the DANA1 expression peaked at 6 days after drought.

4. Can authors comment why there was no ABA related GO category getting enriched in RNA seq data?

5. Even though authors have discussed this, it will be interesting to comment on the parts of DANA1 which are critical for DIP binding (related to Fig.3)

6. From RNA seq, defense GO is enriched in the mutant. It would be interesting to test them for pathogen stress as well in future.

Referee #2:

This is a manuscript that explores the role of a new lncRNA DANA1 in drought response. The authors have done a good job by exploring the different aspects of the physiological drought response. I find the drought tests, water loss, root elongation and stomatal aperture analysis of good quality and relevant for the manuscript. The analysis of the DANA1 lncRNA is also of good quality and extensive with knockouts, complementation lines, and overexpressors.

The link between DANA1 and DIP1 protein is also well documented.

I have several suggestions that could improve the work further:

The list of proteins identified in the triple hybrid screen should be shown in the supplement.

The link between DANA1 and CYP707A1/2 is not well explained I am not sure how did the authors select these targets?

Also while I do appreciate the CYP data a DANA1 ChIPseq would additionally strengthen the work.

In summary, I find this a solid piece of work.

Referee #3:

In this manuscript, the authors investigated the function of the drought-induced long intergenic noncoding RNA DANA1 in Arabidopsis. It was shown that DANA1 interacts with the L1p/L10e family member protein DANA1 INTERACTING PROTEIN 1 (DIP1) in the cell nucleus of Arabidopsis. Furthermore, DIP1 alters the histone deacetylase HDA9 binding on CYP707A1 and CYP707A2 loci through direct interaction with PWWP3, a member of the PEAT complex acting via histone deacetylation, thereby repressing CYP707A1 and CYP707A2 expression. It was proposed that DANA1 functions as a positive regulator of drought response and works jointly with the chromatin-related factor DIP1 to modulate the epigenetic reprogramming of the plant transcriptome during the response to drought. This study provides a new perspective on how noncoding RNAs function in epigenetic regulation of gene expression in plants. However, further evidence is needed to support these findings.

Major points:

1. More genetic analysis is required to investigate the interaction among DANA1, DIP1, HDA9 and PWWP3. The authors need to generate double and/or triple mutants to further analyze their interactions and functions. Furthermore, DANA1 can be overexpressed in *dip1/hda9/pwwp3* mutant background or vice versa to analyze their interaction.
2. I suggest that the authors also need to carry RNA-seq of WT and *dana1-2* plants under stress conditions, which will provide more information about the function and activity of DANA1 in stress responses.
3. In Figure 4, pull down assays and BiFC assays were used to analyze the interaction between DIP1 and PWWP3. The authors need to further analyze the interaction between PWWP3 and DIP1 by CoIP assays.
4. PWWP3 is known to belong to the PEAT complex that associates with the histone deacetylase HDA9. Does DIP1 also interact with HDA9? The authors need to analyze this possibility.

Cross-comments from referee 1:

I agree that the DANA1 ChIPseq is not a mandatory requirement. I think the authors might be able to address some of the comments by referee 3 experimentally. For others, they might find a way to answer them in some way.

Cross-comments from referee 2:

I read the reviews and quickly looked at the manuscript to remind myself about it. As for the comments by the other reviewers, I think the Reviewer one has a good point that DANA could show generic ABA misregulation rather than drought-specific phenotypes in that sense it would be nice to address it by checking ABA levels in nonstressed plants or ABA-related phenotypes as described by him or her. As for Reviewer 3 I think point one has merit in making the work more coherent. In that sense that some of the double mutant analyses proposed or *ox* of DANA in one of the mutants will strengthen the coherence of the work, in my view they are not absolutely necessary but would be nice. I personally think the other points go a bit too far in confirming what is already nicely shown by the authors.

Cross-comments from referee 3:

I agree with you that DANA1-ChIPseq data are not required. I think the authors need to address other comments from both reviewers.

Dear Dr. Esther Schnapp,

Thank you very much for the comments on the manuscript (EMBOR-2023-57737-T) entitled “LncRNA *DANA1* promotes histone deacetylation of drought responsive genes in *Arabidopsis*” that we submitted to EMBO reports. The paper has now been revised according to the suggestions, and our point-by-point responses to reviewers' concerns are detailed as follows:

Rely to Referee #1:

1. It is interesting to study the *dana1* mutant as plants with enhanced ABA catabolism. It is obvious (Fig. 5G) that *dana1* mutants have reduced ABA levels under control conditions which increase in response to drought, albeit to a lower level. So, it is worthwhile to discuss that *dana1* may not be having a drought specific phenotype only but a general ABA deficient phenotype. Is drought tolerance just one of many responses as *dana1* mutants might be highly pleiotropic as seen in many epigenetic machinery mutants. In this regard, it will be great to study the *DANA1* expression during different plant developmental (pre-and post-germination) stages. *dana1* mutants can be tested in detail for their germination, stomatal development and transpiration rate and water use efficiency during seedling and adult plant stages before and after PEG or drought treatment.

Response: Based on this suggestion, expressional levels of *DANA1* during pre- and post-germination stages in Col-0 were examined by RT-qPCR at first. *DANA1* mRNA was downregulated immediately after imbibition, and decreasing gradually (Appendix Fig S20A). However, in the plant post-germination stage, the expression of *DANA1* gradually increased (Appendix Fig S20B). In addition, we tested whether *DANA1* plays a role in seed germination. Mutants of *dana1* exhibited insensitive to PEG treatment at the stage of germination (Appendix Fig S19). Then effects of *DANA1* on stomatal aperture index, stomatal conductance, transpiration rate, and water use efficiency in *Arabidopsis* plants before and after drought treatment were studied. We found that stomatal aperture index, stomatal conductance and transpiration rate were

significantly increased in *dana1* mutants, but water use efficiency was significantly decreased in *dana1* mutants (Appendix Fig S21). These results have been added into our revised manuscript as “The ABA content in *DANA1* OE plants was significantly higher than in Col-0 WT (Appendix Fig S17). Both stomatal index and stomatal density were significantly decreased in *DANA1* OE plants compared with WT (Appendix Fig S18), which is consistent with the phenotype of *cyp707a1/cyp707a3* double mutant (Tanaka *et al.*, 2013). It has been known that CYP707A2 plays an essential role in seed dormancy and germination (Kushiro *et al.*, 2004), thus, we tested whether *DANA1* plays a role in seed germination. Mutants of *dana1* exhibited insensitive to PEG treatment at the stage of germination (Appendix Fig S19). Moreover, expressional levels of *DANA1* during pre- and post-germination stages in Col-0 were examined. *DANA1* mRNA was downregulated immediately after imbibition, and decreasing gradually (Appendix Fig S20A). However, in the plant post-germination stage, the expression of *DANA1* gradually increased (Appendix Fig S20B). Then effects of *DANA1* on stomatal aperture index, stomatal conductance, transpiration rate, and water use efficiency in *Arabidopsis* plants before and after drought treatment were studied. We found that stomatal aperture index, stomatal conductance and transpiration rate were significantly increased in *dana1* mutants, but water use efficiency was significantly decreased in *dana1* mutants (Appendix Fig S21).”, “Intrinsic water use efficiency (WUE_i) was calculated with the following formula: $WUE_i = \text{Photosynthesis rate} / \text{transpiration rate}$.” and “Measurements of stomatal index and stomatal density were performed as described previously (Xiao *et al.*, 2021). In brief, the cotyledons of 5-day-old seedlings were analyzed for stomatal index and stomatal density, and the central areas derived from the abaxial cotyledon surface were imaged using an LSM 900 confocal laser scanning microscope (Zeiss).

Stomata and pavement cell numbers were counted with ImageJ software, and stomatal index was calculated by dividing the number of stomata by the total amount of cells (stomata + pavement cells).” (Page 17-18 lines 392-408, Page 23 lines 534-535 and Page 23 lines 536-542).

Appendix Figure S20. Quantitative measurement of the transcript levels of *DANA1* during pre- (A) and post-germination (B) stages.

A The RT-qPCR was performed using dry seed and imbibed seed at the designated time points. For seed imbibition, it was carried out as the previous study (Liu *et al.*, 2009), and seeds were sown and imbibed on filter paper moistened with water at 22°C under continuous light.

B The RT-qPCR was performed using dry seed and seed that were sown on 1/2 MS media, stratified for 3 days at 4°C in the dark and then grown at 22°C with a 16-h light/8-h dark photoperiod at the designated time points. Expression level of *DANA1* in dry seed was designed as 1, and *UBQ3* was used as an internal control.

Appendix Figure S19. Seed germination of Col-0, *dana1-1* and *dana1-2* mutants.

Seeds (30 per genotype) were grown on half MS medium supplemented with 30% (w/v) PEG. Seeds were stratified at 4°C for 3 days, and germinations of seeds were determined when they had grown for 2 days. Seeds were considered as germinated once radicle penetrated the seed coat. Values shown are means \pm SD from three

replicates. Asterisks represent significant differences determined by Student's *t*-test (* $P < 0.05$).

Appendix Figure S21. Stomatal aperture index (A), stomatal conductance (B), transpiration rate (C) and water use efficiency (D) in the rosette leaves of four-week-old Col-0, *dana1-1* and *dana1-2* mutants grown under normal and drought stress conditions.

Data information: Values shown are means \pm SD from three replicates. Asterisks represent significant differences determined by Student's *t*-test (* $P < 0.05$; ** $P < 0.01$; *** $P < 0.001$).

References:

- Kushiro T, Okamoto M, Nakabayashi K, Yamagishi K, Kitamura S, Asami T, Hirai N, Koshiba T, Kamiya Y, Nambara E (2004) The Arabidopsis cytochrome P450 CYP707A encodes ABA 8'-hydroxylases: key enzymes in ABA catabolism. *EMBO J* 23: 1647-56
- Liu Y, Shi L, Ye N, Liu R, Jia W, Zhang J (2009) Nitric oxide-induced rapid decrease of abscisic acid concentration is required in breaking seed dormancy in Arabidopsis. *New Phytol* 183: 1030-1042
- Tanaka Y, Nose T, Jikumaru Y, Kamiya Y (2013) ABA inhibits entry into

stomatal-lineage development in Arabidopsis leaves. *Plant J* 74: 448-457

Xiao C, Guo H, Tang J, Li J, Yao X, Hu H (2021) Expression Pattern and Functional Analyses of Arabidopsis Guard Cell-Enriched GDSL Lipases. *Front Plant Sci* 12: 748543

2. The good test that DANA1 regulates the CYP707 expression, would be to test the ABA levels in DANA1 OE lines. In that respect it will be interesting to check whether OE lines have a *cyp707* mutant like phenotype like Decreased stomatal index and density (Tanaka et al *Plant J* 2013).

Response: We thank the reviewer for this suggestion. The ABA content in *DANA1* OE plants was measured, and we found that it was significantly higher than in Col-0 wild type (Appendix Fig S17). Additionally, both stomatal index and stomatal density were significantly decreased in *DANA1* OE plants compared with WT (Appendix Fig S18), which is consistent with the phenotype of *cyp707a1/cyp707a3* double mutant (Tanaka *et al.*, 2013). These results have been added into our revised manuscript as “The ABA content in *DANA1* OE plants was significantly higher than in Col-0 WT (Appendix Fig S17). Both stomatal index and stomatal density were significantly decreased in *DANA1* OE plants compared with WT (Appendix Fig S18), which is consistent with the phenotype of *cyp707a1/cyp707a3* double mutant (Tanaka *et al.*, 2013).” and “and 10-day-old seedlings of Col-0 and *DANA1* over-expressing lines grown on 1/2 MS agar plates were collected”. (Page 17 lines 392-396 and Page 23 lines 546-547).

Appendix Figure S17. Measurement of ABA contents in seedlings. The ABA contents were measured in 10-day-old Col-0 and *DANA1*-overexpression seedlings.

Data information: Asterisks represent significant differences by Student's t-test (** *P*

< 0.01; *** $P < 0.001$, Student's t -test).

Appendix Figure S18. Both stomatal index (A) and stomatal density (B) are decreased in *DANA1* over-expressing plants.

Data information: Values shown are means \pm SD ($n = 15$). Asterisks represent significant differences determined by Student's t -test (* $P < 0.05$; ** $P < 0.01$; *** $P < 0.001$).

References:

Tanaka Y, Nose T, Jikumaru Y, Kamiya Y (2013) ABA inhibits entry into stomatal-lineage development in *Arabidopsis* leaves. *Plant J* 74: 448-457.

3. Are the drought phenotype in *dana1* and *dip1* mutants ABA dependent? Can authors comment on what regulates the expression of *DANA1* itself during drought treatment. Is there a possibility of a feedback loop regulation. It will be great to test the expression of *DANA1* in ABA signaling mutants like *pyl/pyr*.

Response: According to reviewer's advice, we measured *DANA1* RNA abundance in WT and *pyr1/pyl1/pyl4* triple mutant by RT-qPCR. Expression of *DANA1* in *pyr1/pyl1/pyl4* plants was significantly higher than in Col-0 plants under both control conditions and dehydration, indicating that *DANA1* expression might be negatively regulated by ABA signaling. This result has been added into our revised manuscript as "Given that ABA-dependent stress responses largely depend on endogenous ABA levels in the plant, which fluctuates widely in response to drought stress (Seki *et al.*, 2007), we employed RT-qPCR to measure *DANA1* RNA abundance in the ABA signaling mutant, *pyr1/pyl1/pyl4* triple mutant. Expression of *DANA1* in *pyr1/pyl1/pyl4* plants was significantly higher than in Col-0 plants under both control conditions and dehydration treatment (Appendix Fig S22), indicating that *DANA1*

expression might be negatively regulated by ABA signaling.” (Page 18 lines 410-416).

Appendix Figure S22. Expression of *DANA1* in WT and an ABA-insensitive mutant *pyr1/pyl1/pyl4*. *UBQ3* was used as an internal control.

Data information: Asterisks represent significant differences determined by Student's *t*-test (***) $P < 0.001$).

References:

Seki M, Umezawa T, Urano K, Shinozaki K (2007) Regulatory metabolic networks in drought stress responses. *Curr Opin Plant Biol* 10: 296-302

4. Though authors have nicely tested the histone enrichment in Fig.5, however, it will be great to test the dynamic histone acetylation before and after the drought treatment around these loci.

Response: We thank the reviewer for this suggestion. H3K9ac and H3K27ac profiles for both *CYP707A1* and *CYP707A2* loci in Col-0, *dana1-2* and *dip1-1* plants under dehydration treatment were investigated, and we found that dehydration treatment causes the decreases of H3K9ac and H3K27ac levels and increase of HDA9 enrichment at both loci (Fig 5A-D). These results have been added into our revised manuscript as “Moreover, H3K9ac and H3K27ac profiles for both *CYP707A1* and *CYP707A2* loci in Col-0, *dana1-2* and *dip1-1* plants under dehydration treatment were investigated, and we found that dehydration treatment causes the decreases of H3K9ac and H3K27ac levels and increase of HDA9 enrichment at both loci (Fig 5A-D).” (Page 14 lines 319-322).

Figure 5. H3K9ac and H3K27ac marks along the *CYP707A1* and *CYP707A2* loci are modulated by *DANA1* regulation of HDA9 deposition.

A Gene structure of *CYP707A1* and *CYP707A2*, indicating exons (boxes) and introns (lines). The locations of the gene regions analyzed by ChIRP-qPCR and ChIP-qPCR are marked.

B The presence of H3K9ac was measured on *CYP707A1* and *CYP707A2* by ChIP-qPCR in WT, *dana1-2* and *dip1-1* plants.

C The presence of H3K27ac was measured on *CYP707A1* and *CYP707A2* by ChIP-qPCR in WT, *dana1-2* and *dip1-1* plants.

D The HDA9 enrichment at *CYP707A1* and *CYP707A2* chromatin depends on *DANA1* and DIP1.

E Even and odd probes successfully retrieve *DANA1* RNA. LacZ probes are used as negative control.

F *DANA1* association to DNA of *CYP707A1* and *CYP707A2* by ChIRP-qPCR in WT plants. Data from ChIRP-qPCR are represented relative to the background level of DNA precipitation (*PP2A*), and the *TA3* locus was used as the negative control (Li *et al.*, 2016).

G ABA content was measured in the rosette leaves of 3-week-old Col-0, *dana1* mutant, *dip1* mutant plants under 10 days of drought stress or not ($n = 3$, n refers to biological replicates).

Data information: Values shown are means \pm SD from three biological replicates. Asterisks represent significant differences by Student's *t*-test (* $P < 0.05$; ** $P < 0.01$; *** $P < 0.001$).

5. As DIP is a ribosome machinery interacting protein, do the authors check the effect of DIP mutation on translation of the proteins. Can the drought phenotype in the *dip* mutant be partly because of translation impairments in the mutant. As the ribosomal machinery is localized in the cytoplasm, do the authors check the localization of DIP alone.

Response: The subcellular localization of the DIP1 was analyzed using *UBQ10::DIP1-GFP* transgenic lines, and we found that DIP1-GFP protein resides in the nucleus (Appendix Fig S6E and F). Moreover, the observation that the subcellular localization of DIP1 is in the nucleus is also supported by using the *DIP1* complementation line (Figure R2). These results together indicate that DIP1 is not a canonical ribosome component, thus we do not check the effect of DIP1 mutation on protein translation in this study.

Appendix Figure S6. Drought-tolerant phenotype of *DIP1* over-expressing plants.

A, B Detection on the transcript levels (A) and protein levels (B) of *DIP1* in *DIP1* over-expressing transgenic lines. *DIP1* OE-3 and *DIP1* OE-5 are Col-0 plants transformed with *UBQ10: DIP1-GFP*. *DIP1* OE-5 has been used in RIP assay presented in Fig. 3D. ACTIN is shown as a loading control.

C The *DIP1* over-expressing lines are insensitive to PEG treatment. Scale bars = 1 cm.

D Root length and fresh weight of seedlings shown in (C).

E Immunoblot analyses showing the nucleus and cytoplasmic distributions of *DIP1*-GFP protein.

F Subcellular localization of *DIP1*-GFP in rosette leaves of 3-week-old *DIP1* OE-5 plants. Scale bar = 20 μ m.

G Drought-tolerance assay. Col-0, *DIP1* OE-3 and *DIP1* OE-5 plants grown under

normal growth conditions for three weeks were subjected to drought stress for eighteen days and then rewatered for three days. Scale bars = 3 cm.

H Survival rate after drought treatment ($n = 3$ biological replates).

I Water loss in detached leaves of three-week-old Col-0, *DIP1* OE-3 and *DIP1* OE-5 plants.

Data information: Values shown are means \pm SD from three biological replicates. Asterisks represent significant differences determined by Student's *t*-test (* $P < 0.05$; ** $P < 0.01$; *** $P < 0.001$).

Figure R2. Subcellular localization analyses of DIP1-GFP in the 5-day-old *DIP1* complementation line. The *dip1-1* was applied for producing the *DIP1* complementation line. Scale bar = 20 μm .

6. The authors should introduce and discuss the ABA catabolism and CYP707 mutants phenotypes in detail.

Response: As per the reviewer's advice, sentences regarding ABA catabolism and CYP707 mutant's phenotype have been added into our revised manuscript as "The phytohormone abscisic acid (ABA) is the major signaling molecule in plant responses to drought stress. ABA content increases when a land plant is subjected to drought stress, and it rapidly decreases when the land plant is recovering from drought. ABA content in plants is modulated by the balance between its biosynthesis and catabolism, and ABA is catabolized via two pathways, hydroxylation and glucose conjugation (Cutler & Krochdo, 1999; Nambara & Marion-Poll, 2005). The 8'-hydroxylation is thought to be the predominant pathway for ABA catabolism, which is catalyzed by a cytochrome P450 monooxygenase encoded by *CYP707As* (Kushiro *et al.*, 2004; Saito *et al.*, 2004)." (Page 4-5 lines 80-88), and please also see the response to question 1 from Reviewer 1.

References:

Cutler AJ, Krochko JE (1999) Formation and breakdown of ABA. *Trends Plant Sci* 4: 472-478

Kushiro T, Okamoto M, Nakabayashi K, Yamagishi K, Kitamura S, Asami T, Hirai N, Koshiha T, Kamiya Y, Nambara E (2004) The Arabidopsis cytochrome P450 CYP707A encodes ABA 8'-hydroxylases: key enzymes in ABA catabolism. *EMBO J* 23: 1647-56

Nambara E, Marion-Poll A (2005) Abscisic acid biosynthesis and catabolism. *Annu Rev Plant Biol* 56: 165-85

Saito S, Hirai N, Matsumoto C, Ohigashi H, Ohta D, Sakata K, Mizutani M (2004) Arabidopsis CYP707As encode (+)-abscisic acid 8'-hydroxylase, a key enzyme in the oxidative catabolism of abscisic acid. *Plant Physiol* 134: 1439-49

7. Please explain in detail how the PEG plates were made in Methods section.

Response: We thank the reviewer for pointing this out. A description regarding how to prepare the PEG plates has been added into our revised manuscript as “The PEG-infused plates were prepared as described by Verslues *et al.*, (2006). An overlay solution containing PEG was poured over 1/2 MS agar plates, and PEG was allowed to diffuse into the medium.” (Page 21 lines 499-502).

References:

Verslues PE, Agarwal M, Katiyar-Agarwal S, Zhu J, Zhu JK (2006) Methods and concepts in quantifying resistance to drought, salt and freezing, abiotic stresses that affect plant water status. *Plant J* 45: 523-539.

8. Why authors choose 10 d time point for ChIPs while the DANA1 expression peaked at 6 days after drought.

Response: With regard to choosing 10-day-old seedlings for ChIP experiments, they are convenient for collecting and dehydration treatment. In addition, the ChIP analyses on genetic materials under control and dehydration treatment conditions support that *DANA1* and *DIP1* together regulate drought response in *Arabidopsis*.

9. Can authors comment why there was no ABA related GO category getting enriched in RNA seq data?

Response: We thank the reviewer for pointing this out. Appendix Fig S4B only showed top 10 most significantly over-represented GO term for the differentially expressed genes (DEGs). We have replaced Appendix Fig S4B to Dataset EV2

containing all statistically significant (FDR < 0.05) GO term, which clearly showed that genes involved in ABA related GO terms, such as “response to ABA”, “response to hormone” and “response abiotic stimulus”, were significantly over-represented in DEGs. We have replaced “Gene Ontology (GO) enrichment analysis for biological process of the differentially expressed genes (DEGs) indicated that genes involved in “Defense response”, “Response to stress”, “Response to chitin”, “Response to organonitrogen compound” and “Response to stimulus” were most significantly over-represented (Appendix Fig S4B).” to “Gene Ontology (GO) enrichment analysis for biological process of the differentially expressed genes (DEGs) indicated that genes involved in biological processes of “Response to ABA”, “Response to hormone”, “Response to abiotic stimulus”, “Defense response” and so on were significantly over-represented (Dataset EV2).” in our revised manuscript. (Page 9 lines 202-205).

10. Even though authors have discussed this, it will be interesting to comment on the parts of *DANA1* which are critical for DIP binding (related to Fig.3)

Response: Based on this suggestion, we used the *RNAfold* service to analyze the RNA secondary structure of *DANA1*, and then divided it into three fragments (*DANA1*-L1, *DANA1*-L2, and *DANA1*-L3) depending on its predicted structure (Appendix Fig S26A). Then a yeast three-hybrid assay was performed on the full-length *DANA1* and its three fragments, and we found that both the full-length *DANA1* and *DANA1*-L1 can interact with DIP1 in yeast cells (Appendix Fig S26B), indicating that the RNA fragment between nucleotides 1 and 175 of *DANA1* is important for the interaction between *DANA1* and DIP1. These results have been added into our revised manuscript as “We used the *RNAfold* service to analyze the RNA secondary structure of *DANA1*, and then divided it into three fragments (*DANA1*-L1, *DANA1*-L2, and *DANA1*-L3) depending on its predicted structure (Appendix Fig S26A). Then a yeast three-hybrid assay was performed on the full-length *DANA1* and its three fragments, and we found that both the full-length *DANA1* and *DANA1*-L1 can interact with DIP1 in yeast cells (Appendix Fig S26B),

suggesting that the RNA fragment between nucleotides 1 and 175 of *DANA1* is important for the interaction between *DANA1* and DIP1.” (Page 19-20 lines 453-460).

Appendix Figure S26. *DANA1-L1* interacts with DIP1 in yeast cells.

A Secondary structure of the *DANA1* predicted with *RNAfold*.

B Tests of *DANA1*-DIP1 interactions by yeast three-hybrid assays.

11. From RNA seq, defense GO is enriched in the mutant. It would be interesting to test them for pathogen stress as well in future.

Response: As per the reviewer’s advice, we have added a sentence regarding the role of *DANA1* in plant responses to pathogens into Discussion part as “In addition, considering that genes involved in “Defense response” were significantly over-represented based on GO enrichment analysis of the DEGs (Dataset EV2), therefore, it will be interesting to study the role of *DANA1* in plant responses to pathogens henceforth.” (Page 20 lines 464-467).

Rely to Referee #2:

1. The list of proteins identified in the triple hybrid screen should be shown in the supplement.

Response: Based on this suggestion, the list of proteins identified in our yeast three-hybrid assay has been added into our revised manuscript as Appendix Table S2.

Appendix Table S2. The list of *DANA1*-interacting proteins identified by a yeast three-hybrid assay.

Number	Gene ID	Gene description
1	AT1G06380	Ribosomal protein L1p/L10e family (DIP1)
2	AT1G49650	alpha/beta-Hydrolases superfamily protein
3	AT1G58470	Encodes an mRNA-binding protein that contains two RNA recognition motifs (RRMs) and is expressed in proliferating tissues
4	AT3G12470	Polynucleotidyl transferase, ribonuclease H-like superfamily protein
5	AT3G15080	Polynucleotidyl transferase, ribonuclease H-like superfamily protein
6	AT3G07050	Arabidopsis NSN1 encodes a nucleolar GTP- binding protein and is required for maintenance of inflorescence meristem identity and floral organ development
7	AT3G54770	Encodes a putative RNA binding protein that is localized in the nucleus and affects ABA-regulated seed germination of Arabidopsis
8	AT4G04890	Arabidopsis thaliana protodermal factor 2 (PDF2)
9	AT5G14610	DEAD box RNA helicase family protein
10	AT5G46250	AtLARP6a, La related protein 6a
11	AT5G52470	ATFIB1
12	AT5G60980	Nuclear transport factor 2 (NTF2) family protein with RNA binding (RRM-RBD-RNP motifs) domain-containing protein

2. The link between *DANA1* and *CYP7007A1/2* is not well explained I am not sure how did the authors select these targets?

Response: We thank the reviewer for pointing this out. Actually, *CYP707A1/A2* are identified as the direct target of *DANA1* until we found that DIP1 could interact with PWWP3. Because that PWWP3, as a component of the PEAT complex that associates with HDA9 histone deacetylase (Tan *et al.*, 2018), and HDA9 is reported to regulate drought response through modulating expression of *CYP707A1* and *CYP707A2* in *Arabidopsis* (Baek *et al.*, 2020). Moreover, DIP1 binds to *CYP707A1* and *CYP707A2* but not *CYP707A4*, *UGT71B1*, *MYB44* and *NTL6* loci (Appendix Fig S15A-C), further supporting this conclusion that *CYP707A1* and *CYP707A2* are direct targets of *DANA1*-DIP1 complex. For the better linking *DANA1* to *CYP707A1/A2*, we have

readjusted the order of Appendix Figure S10F-H in our previous manuscript to Appendix Figure S15A-C in our revised manuscript, and added the description of *CYP707A* function into Introduction of our revised manuscript, and please also see the response to question 6 from Reviewer 1.

References:

Baek D, Shin G, Kim MC, Shen M, Lee SY, Yun DJ (2020) Histone Deacetylase HDA9 With ABI4 Contributes to Abscisic Acid Homeostasis in Drought Stress Response. *Front Plant Sci* 11: 143

Tan LM, Zhang CJ, Hou XM, Shao CR, Lu YJ, Zhou JX, Li YQ, Li L, Chen S, He XJ (2018) The PEAT protein complexes are required for histone deacetylation and heterochromatin silencing. *EMBO J* 37: e98770

3. Also while I do appreciate the CYP data a DANA1 ChIPseq would additionally strengthen the work.

Response: We agree with reviewer's perspective that the examination of *DANA1* binding to *CYP707A1/2* loci by ChIRP-seq will be helpful for better understanding *DANA1*-mediated drought stress in plants, which will be performed in our future work. In addition, we investigated the *DANA1* association to DNA of *CYP707A1* and *CYP707A2* in Col-0 plants under dehydration treatment, and found that dehydration treatment causes the increase of *DANA1* binding at both loci (Fig 5F), which further supports our conclusion that *DANA1* regulates plant drought response by modulating *CYP707A1* and *CYP707A2* expression. This result has been added into our revised manuscript as "By performing ChIRP-qPCR, we confirmed that *DANA1* RNA physically associates with DNA from the *CYP707A1* and *CYP707A2* loci, and dehydration treatment causes the increase of *DANA1* binding at both loci (Fig 5F)." (Page 14 lines 327-329).

Figure 5. H3K9ac and H3K27ac marks along the *CYP707A1* and *CYP707A2* loci are modulated by *DANA1* regulation of HDA9 deposition.

A Gene structure of *CYP707A1* and *CYP707A2*, indicating exons (boxes) and introns (lines). The locations of the gene regions analyzed by ChIRP-qPCR and ChIP-qPCR are marked.

B The presence of H3K9ac was measured on *CYP707A1* and *CYP707A2* by ChIP-qPCR in WT, *dana1-2* and *dip1-1* plants.

C The presence of H3K27ac was measured on *CYP707A1* and *CYP707A2* by ChIP-qPCR in WT, *dana1-2* and *dip1-1* plants.

D The HDA9 enrichment at *CYP707A1* and *CYP707A2* chromatin depends on *DANA1* and DIP1.

E Even and odd probes successfully retrieve *DANA1* RNA. LacZ probes are used as negative control.

F *DANA1* association to DNA of *CYP707A1* and *CYP707A2* by ChIRP-qPCR in WT plants. Data from ChIRP-qPCR are represented relative to the background level of DNA precipitation (*PP2A*), and the *TA3* locus was used as the negative control (Li *et al.*, 2016).

G ABA content was measured in the rosette leaves of 3-week-old Col-0, *dana1* mutant, *dip1* mutant plants under 10 days of drought stress or not ($n = 3$, n refers to biological replicates).

Data information: Values shown are means \pm SD from three biological replicates. Asterisks represent significant differences by Student's *t*-test (* $P < 0.05$; ** $P < 0.01$; *** $P < 0.001$).

In summary, I find this a solid piece of work.

Response: We thank the reviewer for the comments on our manuscript.

Rely to Referee #3:

1. More genetic analysis is required to investigate the interaction among *DANA1*, *DIP1*, *HDA9* and *PWWP3*. The authors need to generate double and/or triple mutants to further analyze their interactions and functions. Furthermore, *DANA1* can be overexpressed in *dip1/hda9/pwwp3* mutant background or vice versa to analyze their interaction.

Response: Based on this suggestion, we produced the *dana1-2/dip1-1* double mutant and *DANA1* OE-15 (*dip1-1*) by crossing *dana1-2* with *dip1-1*, *DANA1* OE-15 and *dip1-1*, respectively. We then performed the drought-tolerance assay and water loss assay on *dana1-2/dip1-1* and *DANA1* OE-15 (*dip1-1*) plants. The *dana1-2/dip1-1* double mutant, *DANA1* OE-15 (*dip1-1*) and *DIP1* OE-5 (*dana1-2*) exhibited drought-sensitive phenotypes, which were consistent with the drought-sensitive phenotypes of the *dana1* and *dip1* mutants (Appendix Fig S10A and B, S10D and E, S10G and H). Increased water loss was found in detached leaves of *dana1-2/dip1-1* double mutant, *DANA1* OE-15 (*dip1-1*) and *DIP1* OE-5 (*dana1-2*), which are in agreement with their drought-sensitive phenotypes (Appendix Fig S10C, F and I).

These results further support our conclusion that *DANA1* regulates drought response in *Arabidopsis* via *DIP1*. In addition, these results have been added to our revised manuscript as “We then investigated the genetic interaction between *DANA1* and *DIP1* in the context of drought tolerance by generating *dana1-2/dip1-1* double mutant, *DANA1* over-expressing lines in *dip1-1* (*DANA1* OE-15 (*dip1-1*)) and *DIP1* over-expressing lines in *dana1-2* (*DIP1* OE-5 (*dana1-2*)). The *dana1-2/dip1-1* double mutant, *DANA1* OE-15 (*dip1-1*) and *DIP1* OE-5 (*dana1-2*) were drought-sensitive, consistent with the drought-sensitive phenotypes of both *dana1* and *dip1* mutants (Appendix Fig S10A and B, S10D and E, S10G and H). Increased water loss was found in detached leaves of *dana1-2/dip1-1* double mutant, *DANA1* OE-15 (*dip1-1*) and *DIP1* OE-5 (*dana1-2*), which are in agreement with their drought-sensitive phenotypes (Appendix Fig S10C, F and I).” and “For *dana1-2/dip1-1*, it is obtained by crossing *dana1-2* with *dip1-1*. For *DANA1* OE (*dip1-1*), it is obtained by crossing *DANA1* OE-15 with *dip1-1*.” (Page 12-13 lines 273-282 and Page 21 lines 491-492).

Appendix Figure S10. Genetic interaction of *DANA1* with *DIP1*.

A Morphology of seedlings before and after drought stress treatment. Three-week-old Col-0, *dana1-2*, *dip1-1* and *dana1-2/dip1-1* plants were subjected to drought stress for sixteen days and then rewatered for five days. Scale bars = 3 cm.

B Survival rate after drought treatment ($n = 3$ biological replates).

C Water loss in detached leaves of three-week-old Col-0, *dana1-2*, *dip1-1* and *dana1-2/dip1-1* plants ($n = 3$ biological replicates, each replicate contains five fully expanded leaves).

D Morphology of seedlings before and after drought stress treatment. Three-week-old Col-0, *DANA1* OE-15, *dip1-1* and *DANA1* OE-15 (*dip1-1*) plants were subjected to drought stress for sixteen days and then rewatered for five days. Scale bars = 3 cm.

E Survival rate after drought treatment ($n = 3$ biological replates).

F Water loss in detached leaves of three-week-old Col-0, *DANA1* OE-15, *dip1-1* and *DANA1* OE-15 (*dip1-1*) plants ($n = 3$ biological replicates, each replicate contains five fully expanded leaves).

G Morphology of seedlings before and after drought stress treatment. Three-week-old Col-0, *dana1-2*, *DIP1* OE-5 and *DIP1* OE-5 (*dana1-2*) plants were subjected to drought stress for sixteen days and then rewatered for five days. Scale bars = 3 cm.

H Survival rate after drought treatment ($n = 3$ biological replates).

I Water loss in detached leaves of three-week-old Col-0, *dana1-2*, *DIP1* OE-5 and *DIP1* OE-5 (*dana1-2*) plants ($n = 3$ biological replicates, each replicate contains five fully expanded leaves).

Data information: Values shown are means \pm SD from three biological replicates. Asterisks represent significant differences by Student's *t*-test (* $P < 0.05$; ** $P < 0.01$; *** $P < 0.001$).

2. I suggest that the authors also need to carry RNA-seq of WT and *dana1-2* plants under stress conditions, which will provide more information about the function and activity of *DANA1* in stress responses.

Response: We thank the reviewer for this suggestion. This study focuses on *DANA1*-mediated drought response by modulating *CYP707A1* and *CYP707A2* expression, and additional function and activity of *DANA1* in stress responses will be the studied in future. Please also see the responses to questions 4 and 11 from Reviewer 1 and the response to question 3 from Reviewer 2.

3. In Figure 4, pull down assays and BiFC assays were used to analyze the interaction between *DIP1* and *PWWP3*. The authors need to further analyze the interaction between *PWWP3* and *DIP1* by CoIP assays.

Response: A co-immunoprecipitation (Co-IP) assay was performed to analyze the interaction between *PWWP3* and *DIP1* according to reviewer's advice, and we found that *PWWP3* protein could interact with *DIP1* protein (Fig 4D). This result has been

added into our revised manuscript as “In addition, the interaction between DIP1 and PWWP3 was also confirmed *in vivo* by using a BiFC assay in *Arabidopsis* protoplast cells and a co-immunoprecipitation (Co-IP) assay in *Nicotiana benthamiana* (Fig 4C and D).” and “Co-IP assay was conducted as previously described (Yang *et al.*, 2023). Protein extracts from infiltrated leaves were immunoprecipitated with anti-GFP (Roche). After washing the beads, the immunoprecipitated proteins were resolved by SDS-PAGE and detected by immunoblotting with anti-GFP (BBI life sciences) or anti-MYC (Abmart). For Co-IP assay performed with infiltrated leaves, agrobacteria harbouring *UBQ10: DIP1-GFP*, *UBQ10: GFP* and *35S: PWWP3-MYC* constructs described above, were transiently expressed in leaves of *N. benthamiana*.” (Page 13 lines 293-295 and Page 26-27 lines 628-634).

Figure 4. PWWP3 mutants are more sensitive to drought stress

A DIP1 interaction with a C-terminal region of PWWP3 (aa 361-645) in yeast cells.

B *In vitro* pull-down assay of MBP-DIP1 and GST-PWWP3.

C BiFC (Bimolecular Fluorescence Complementation) assay of the interaction between DIP1 and PWWP3 in *Arabidopsis* protoplast cells. *Arabidopsis* mesophyll protoplasts were transformed with different plasmid pairs as follows. DIP1-cYFP/PWWP3-nYFP: co-transformation of DIP1 fused C terminus of YFP and PWWP3 fused N terminus of YFP; DIP1-cYFP/nYFP: co-transformation of DIP1 fused C terminus of YFP and N terminus of YFP; cYFP/PWWP3-nYFP: co-transformation of C terminus of YFP and PWWP3 fused N terminus of YFP. Scale bars = 10 μ m.

D Co-immunoprecipitation (Co-IP) assay of DIP1 and PWWP3. *PWWP3-MYC* was cotransformed with *DIP1-GFP* or *GFP* in *Nicotiana benthamiana* leaves. The expressed proteins were immunoprecipitated using an anti-GFP antibody and then detected with anti-MYC antibody.

E *PWWP3* mutants are sensitive to PEG treatment. Scale bars = 1 cm.

F Root length and fresh weight of seedlings shown in (E). Error bars represent standard deviation ($n = 15$, n refers to biological replicates).

G Drought-tolerance assay. Col-0, *pwwp3-1* and *pwwp3-2* plants grown under normal growth conditions for three weeks were subjected to drought stress for thirteen days and then re-watered for five days. Scale bars = 3 cm.

H Survival rate after drought treatment ($n = 3$ biological replicates, n refers to biological replates).

I Water loss in detached leaves of three-week-old Col-0, *pwwp3-1* and *pwwp3-2* plants.

Data information: Values shown are means \pm SD from three biological replicates. Asterisks represent significant differences determined by Student's *t*-test (* $P < 0.05$; ** $P < 0.01$; *** $P < 0.001$).

References:

Yang J, He R, Qu Z, Gu J, Jiang L, Zhan X, Gao Y, Adelson DL, Li S, Wang ZY, Zhu Y, Wang D (2023) Long noncoding RNA ARTA controls ABA response through MYB7 nuclear trafficking in *Arabidopsis*. *Dev Cell* 58: 1206-1217 e4

4. PWWP3 is known to belong to the PEAT complex that associates with the histone deacetylase HDA9. Does DIP1 also interact with HDA9? The authors need to analyze this possibility.

Response: As per the reviewer's advice, we investigated the interaction between DIP1 and HDA9 by yeast two-hybrid assay, and found that DIP1 could not directly interact with HDA9 in yeast cells (Appendix Fig S24). This result has been added into our revised manuscript as "Then, the interaction between DIP1 and HDA9 was investigated by yeast two-hybrid assay, and we found that DIP1 could not directly interact with HDA9 in yeast cells (Appendix Fig S24)." (Page 18 lines 421-423).

Appendix Figure S24. Tests of DIP1-HDA9 interaction by yeast two-hybrid assays. Full-length HDA9 protein was fused with the GAL4 activation domain (AD).

We believe that we have satisfactorily addressed all the questions raised by the reviewer, and we hope that our revised paper is now suitable for publication in EMBO reports.

Sincerely yours

Dong Wang

Dear Dr. Wang,

Thank you for the submission of your revised manuscript. We have now received the enclosed reports from the referees and I am happy to say that all support its publication now. Only a few editorial requests will need to be addressed before we can proceed with the official acceptance of your manuscript:

- Please rename the "Data Availability Section" and place it before the Acknowledgements.
- Please remove the author credits from the ms file. All credits need to be entered into our online submission system.
- Please upload all figures as individual files. The figure legends need to be removed from the figures and added to the manuscript file.
- Please also upload the Source Data as one folder per figure.
- In Appendix Fig S6 - B Anti Actin there might be a possible splice site. Please send us the source data for this panel.
- The data callout in the text for the GSE113677 data citation does not include "Data ref:" as a prefix. Please correct.

I would like to suggest some small changes to the abstract. Please let me know whether you agree with the following and whether all is correct:

Although many long noncoding RNAs have been discovered in plants, little is known about their biological function and mode of action. Here we show that the drought-induced long intergenic noncoding RNA DANA1 interacts with the L1p/L10e family member protein DANA1 INTERACTING PROTEIN 1 (DIP1) in the cell nucleus of Arabidopsis, and both DANA1 and DIP1 promote plant drought resistance. DANA1 and DIP1 increase histone deacetylase HDA9 binding to the CYP707A1 and CYP707A2 loci. DIP1 further interacts with PWWP3, a member of the PEAT complex that associates with HDA9 and has histone deacetylase activity. Mutation of DANA1 enhances CYP707A1 and CYP707A2 acetylation and expression resulting in impaired drought tolerance, in agreement with *dip1* and *pwwp3* mutant phenotypes. Our results demonstrate that DANA1 is a positive regulator of drought response and that DANA1 works jointly with the novel chromatin-related factor DIP1 on epigenetic reprogramming of the plant transcriptome during the response to drought.

EMBO press papers are accompanied online by A) a short (1-2 sentences) summary of the findings and their significance, B) 2-3 bullet points highlighting key results and C) a synopsis image that is exactly 550 pixels wide and 200-600 pixels high (the height is variable). You can either show a model or key data in the synopsis image. Please note that text needs to be readable at the final size. Please send us this information along with the final manuscript.

Referee #1:

The revised manuscript addressed many of the comments of the reviewers. Importantly, the additional data added to the revised version of the manuscript support the claims and conclusions made by the authors. I therefore support the publication of this important manuscript.

Referee #2:

All my points were answered. I am satisfied with the review.

Referee #3:

The authors have adequately addressed most of my concerns in the revised manuscript.

All editorial and formatting issues were resolved by the authors.

Dr. Dong Wang
Nanchang University
College of Life Science
Nanchang, Jiangxi 330031
China

Dear Dr. Wang,

I am very pleased to accept your manuscript for publication in the next available issue of EMBO reports. Thank you for your contribution to our journal.

Yours sincerely,
